# Gradient design of imprinted anode for stable Zn-ion batteries

Qinghe Cao[1,2,3,4], Yong Gao[1,2,4], Jie Pu[1,2,4], Xin Zhao[1], Yuxuan Wang[1], Jipeng Chen[1] & Cao Guan [1,2] ✉

Achieving long-term stable zinc anodes at high currents/capacities remains a great challenge for practical rechargeable zinc-ion batteries. Herein, we report an imprinted gradient zinc electrode that integrates gradient conductivity and hydrophilicity for long-term dendrite-free zinc-ion batteries. The gradient design not only effectively prohibits side reactions between the electrolyte and the zinc anode, but also synergistically optimizes electric field distribution, zinc ion flux and local current density, which induces preferentially deposited zinc in the bottom of the microchannels and suppresses dendrite growth even under high current densities/capacities. As a result, the imprinted gradient zinc anode can be stably cycled for 200 h at a high current density/capacity of 10 mA cm$^{-2}$/10 mAh cm$^{-2}$, with a high cumulative capacity of 1000 mAh cm$^{-2}$, which outperforms the none-gradient counterparts and bare zinc. The imprinted gradient design can be easily scaled up, and a high-performance large-area pouch cell (4*5 cm$^2$) is also demonstrated.

Lithium-ion batteries have achieved remarkable success over the past decades, but serious problems such as scarcity of resources, high prices, and safety risks greatly limit their future applications[1–4]. As a promising reliable alternative, rechargeable aqueous zinc-ion batteries have multiple advantages by using Zn anodes, such as high safety, rich natural resources, and high theoretical capacity (820 mAh g$^{-1}$ and 5855 mAh cm$^{-3}$)[5–7]. However, Zn anodes suffer from poor plating/stripping reversibility and notorious dendrite growth, which results in unsatisfactory cycling stability[8–12]. In addition, Zn anodes also face problems by side reactions and corrosions in aqueous electrolytes[13–17]. Such problems would greatly reduce the coulombic efficiency (CE) and capacity, and sharp dendrites can be easily formed and cause battery failure.

To solve the above problems, an effective way is to construct artificial protective layers on the Zn anode. Materials such as ZnS[18], ZnF$_2$[19], Sn[20], PVB[21], and MXene[22] have been reported effective in suppressing side reactions with enhanced stability[23–26]. However, when the local electric field becomes large and bumps appear on the electrode surface, the rapid growth of dendrites (hot-spot effect) can be still

observed[27–30], especially at high current/capacity, thus they only work well under low current density/capacity. Constructing 3D Zn anodes is another promising approach for stable Zn anodes. The 3D structure effectively increases the specific surface area for more reaction sites and reduces the local electric field strength for uniform Zn deposition[31–36]. For example, Chen et al. prepared an AgNPs@CC electrode by using commercial carbon cloth as the 3D scaffold, which achieved good Zn deposition behavior with long cycling stability[37]. Zhang et al. proposed a 3D Ni–Zn electrode with a multi-channel lattice structure and super-hydrophilic surface, which effectively ameliorated the electric field distribution and induced uniformly deposited Zn[31]. In another work, a 3D Zn anode was constructed with microholes using ITO templates, which also achieved good spatial-selection deposition of Zn[33]. Despite such 3D design, the side reactions and Zn dendrite growth still occur on the top surface that facing the separator, which brings about short-circuit problems after long-term repeated cycling. As another effective method, the design of a 3D Zn anode with a gradient structure can improve the local charge transport kinetics and optimize the Zn deposition process[38–43]. For example, Shen et al.

[1]Institute of Flexible Electronics, Northwestern Polytechnical University, Xi'an 710072, China. [2]Key laboratory of Flexible Electronics of Zhejiang Province, Ningbo Institute of Northwestern Polytechnical University, 218 Qingyi Road, Ningbo 315103, China. [3]Department of Materials Science and Engineering, National University of Singapore, Singapore 117576, Singapore. [4]These authors contributed equally: Qinghe Cao, Yong Gao, Jie Pu. ✉e-mail: iamcguan@nwpu.edu.cn

prepared a 3D gradient Zn anode with Cu foam at the bottom, Ni foam in the middle, and NiO coating on the top, which created a gradual increase in Zn/Zn²⁺ reaction resistance from the bottom to the top[44]. The gradient Zn anode effectively avoided the growth of dendrites on the top surface and showed stable cycling for 250 h at 3 mA cm⁻². However, since the gradient electrode used metal foams as the scaffold, the cross-scale variations in the architecture and the non-uniform micro/nanopores could disorder the Zn²⁺ ions diffusion and slow down the charge transfer. In addition, the final Zn anode requires a further Zn deposition process, which not only makes the electrode preparation process complicated, but also the non-active foam occupies much weight thus sacrificing the energy density. Therefore, it will be of great interest to develop new design strategies for Zn anodes that can well control Zn deposition and suppress side reactions, ultimately achieving stable cycling performance at high current density/capacity (≥5 mA (h) cm⁻²).

Herein, we report a facilely imprinted gradient Zn anode (polyvinylidene difluoride-Sn@Zn, noted as PVDF-Sn@Zn) that well integrates gradient conductivity and hydrophilicity for long-term stable Zn-ion batteries. Different from the simple design of artificial protective layers, the top hydrophobic and insulating layer of PVDF and the bottom hydrophilic and conductive layer of Sn creates a double-gradient structure. The hydrophobic PVDF layer effectively prevents Zn metal from corrosion in the electrolyte, while the Sn layer with a high redox potential (Sn²⁺/Sn, −0.136 V vs SHE) inhibits the side reactions of Zn. In addition, the gradient conductivity effectively induces electric field distribution, Zn²⁺ ion flux, and local current density toward the bottom of the microchannels, thus achieving desired bottom-up deposition behavior for Zn metal. Therefore, not only controllable and uniform Zn deposition is achieved, but also the possible short-circuit from top dendrite growth is prevented. As a result, the PVDF-Sn@Zn gradient electrode shows stable cycling of 1200 h at a low current density/capacity of 1 mA cm⁻²/1 mAh cm⁻² and 200 h at a high current density/capacity of 10 mA cm⁻²/10 mAh cm⁻². Moreover, the simple imprinting method with facile gradient design can be easily

scale-up and a high-performance large-area pouch cell (4 cm² × 5 cm²) is also demonstrated.

## Results

### Gradient electrode design

To illustrate the advantages of the gradient design, controlled samples of bare Zn and 3D Zn without gradient structure are constructed and compared. Generally, bare Zn foil undergoes severe corrosion and side reaction in the electrolyte, and the deposited Zn is inhomogeneous, thus Zn dendrite can be easily formed with poor cycling performance (Fig. 1a). 3D Zn anode can better reduce the local electric field strength, thus uniform Zn deposition can be observed in the microchannels. However, side reactions and dendrite growth still occur at the top surface (close to the separator), which brings about significant risks of short circuits (Fig. 1b). In order to solve the above problems, a gradient Zn anode with microchannels that well integrates gradient conductivity and hydrophilicity is proposed (Fig. 1c). In each microchannel, the top surface is a hydrophobic and insulating layer of PVDF and the bottom is a layer of hydrophilic and conductive Sn. Such gradient design has the following merits: Firstly, the hydrophobic PVDF increases the interfacial free energy between the Zn substrate and the electrolyte, and the hydrophilic Sn with high redox potential facilitates the penetration of the electrolyte, both of which contribute to the inhibition of the side reactions of Zn metal with the aqueous electrolyte. Secondly, the insulating PVDF prevents the deposition of Zn on the top surface, while the bottom Sn with good conductivity triggers the electron-ion exchange more easily, and the gradient conductivity effectively induces electric field distribution, Zn²⁺ ion flux and local current density toward the bottom of the microchannels, thus achieving desired bottom-up deposition behavior for Zn metal. Besides, the zincophilic Sn can uniform the Zn nucleation and deposition process, further suppressing the dendrite growth (The zincophilicity gradient is not considered since Zn would favorably deposit at Sn layers). As a result, such gradient Zn electrodes with microchannels would illustrate promising electrochemical properties.

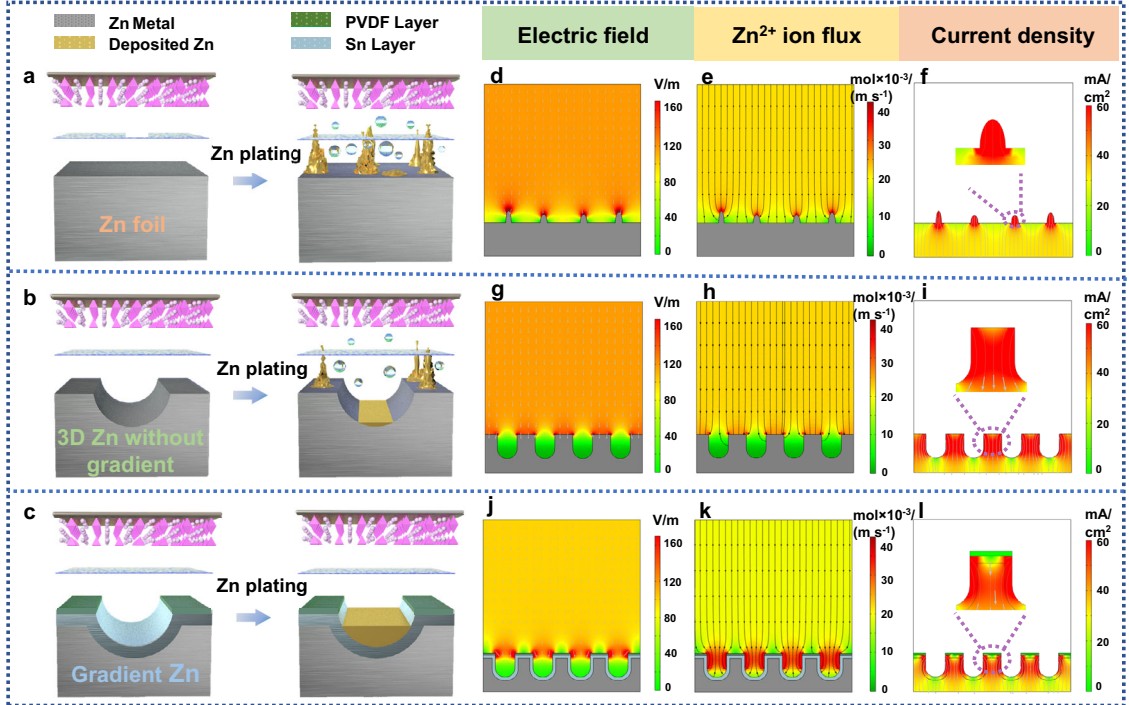

**Fig. 1 | Zn deposition behavior and the finite element simulations of three different electrodes (bare Zn, 3D Zn without gradient, and gradient Zn).** The model of **a** Zn foil, **b** 3D Zn without gradient, and **c** gradient Zn electrodes after Zn deposition. Simulations of electric field distribution, Zn²⁺ ion flux, and current density of **d**–**f** Zn foil, **g**–**i** 3D Zn without gradient, and **j**–**l** gradient Zn electrodes.

The simulations of the electric field distribution, $Zn^{2+}$ ion flux, and current density of the gradient Zn anode are further studied. Bare Zn foil and 3D Zn anode without gradient structure are also modeled for comparison. For bare Zn, the surface roughness would result in the concentrated electric field strength and current density on the bumps, which causes severe polarization. In combination with such inhomogeneous $Zn^{2+}$ ion flux, Zn will aggressively deposit on the bumps and form dendrites (Fig. 1d–f). For 3D Zn (without gradient design), since lower local electric field strength and optimized $Zn^{2+}$ ion flux can be achieved, improved Zn deposition behavior is observed (Fig. 1g, h). However, the current density at the top surface is very high (Fig. 1i), which brings about high risks of side reactions and dendrite growth for short circuits. In contrast, for the gradient electrode, since the big difference in conductivity between the top and the bottom layers, the electric field strength and $Zn^{2+}$ ion flux is more concentrated in the microchannels with values tending to increase at the bottom (Fig. 1j, k). The low current density at the top surface also promotes the Zn deposition preferentially at the bottom of the microchannels (Fig. 1l). Therefore, benefiting from the larger current density, optimized $Zn^{2+}$ ion flux, and electric field at the bottom, the deposited Zn in the gradient electrode is largely moved to the bottom part, which is desirable to prevent top dendrite growth and achieve bottom-up Zn deposition behavior with enhanced cycling stability.

## Preparation and characterization

Based on the above discussion, 3D Zn electrodes with gradient microchannels would express optimized Zn deposition behavior with enhanced electrochemical performance. In this regard, we construct a gradient Zn anode with microchannels (PVDF-Sn@Zn) with the help of the facile imprinting technique (Fig. 2a and Methods). Firstly, Sn decorated Zn electrode (Sn@Zn) is prepared by immersing a piece of Zn foil in an $SnCl_4$ solution, in which Zn spontaneously reduces $Sn^{4+}$ to Sn under the following reaction: $Sn^{4+}+2Zn\rightarrow Sn+2Zn^{2+}$. Subsequently,

the obtained Sn@Zn and a pre-cleaned commercial stainless-steel mesh (SSM, 500-mesh) are passed together through a mechanical roller press, during which the SSM is imprinted on the Sn@Zn. Finally, PVDF is coated on the imprinted Sn@Zn and the final PVDF-Sn@Zn gradient electrode is obtained after removing the SSM template. Noting that in the whole preparation process, no high temperature or complex post-treatment is involved and the SSM template can be simply reused, which effectively simplifies the fabrication process and reduces the cost. Benefiting from the facile and continuous preparation process, the gradient PVDF-Sn@Zn electrode can be also prepared with large size of $10\,cm^2 \times 15\,cm^2$ (Supplementary Fig. 1). From scanning electron microscopy (SEM) images in Fig. 2b–d and Supplementary Fig. 2, the Sn@Zn electrode exhibits good flatness, and the SSM is uniformly imprinted on the surface of Sn@Zn and creates regular microchannel structures. After the PVDF coating process, the uncovered parts by the SSM on the Sn@Zn surface are uniformly coated with PVDF. After removing the SSM template, the parts imprinted by SSM form microchannels (with the size of ~28 μm) and are uniformly coated with the Sn layer, while the other areas of the Sn@Zn are covered by PVDF, thus forming a unique gradient electrode with microchannels (Fig. 2e, f). This gradient structure exhibits void space with an enlarged specific surface area compared to the planar one, thus optimizing the electric field strength on the electrode surface (Supplementary Fig. 3). The gradient electrode is further studied by energy-dispersive spectroscopy (EDS) mapping, where F element is only observed from the areas without SSM imprinting (Fig. 2g–i). From the cross-sectional SEM image with the corresponding mapping results, the thickness of the upper PVDF layer is about 2.4 μm and the Sn layer is about 4 μm (Fig. 2j–m). The characteristic peaks of F and Sn obtained by X-ray photoelectron spectroscopy (XPS) also prove the successful preparation of this gradient electrode (Supplementary Fig. 4). Four-probe resistivity experiment is carried out to clarify the conductivity gradient in the electrode, where the values for PVDF@Zn

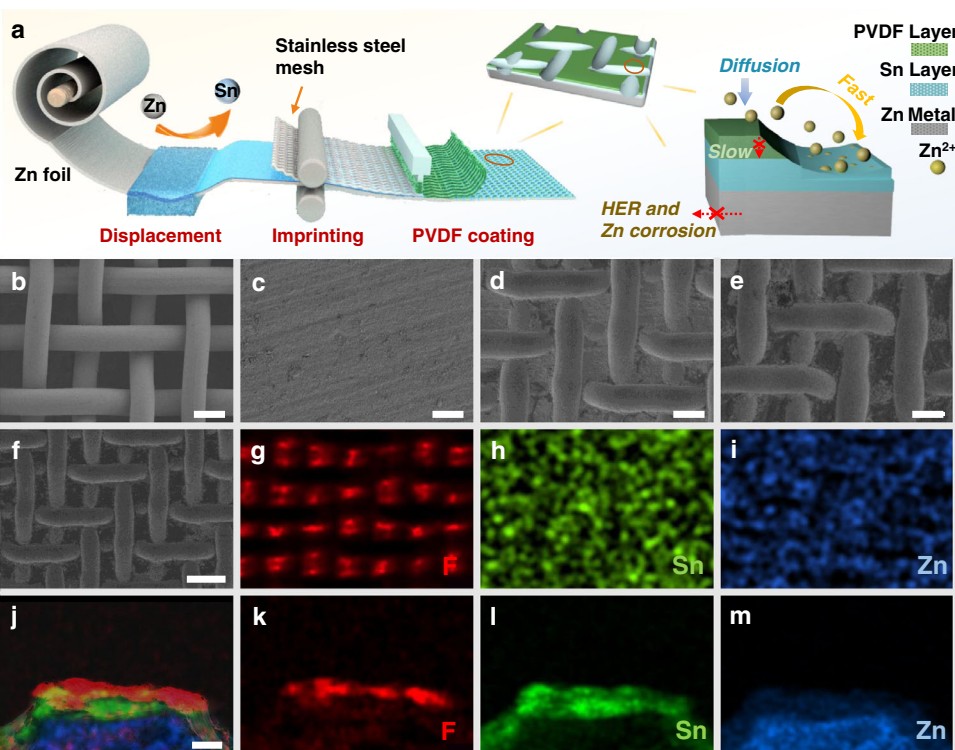

**Fig. 2 | Preparation and characterization of PVDF-Sn@Zn gradient electrode. a** Schematic illustration of the fabrication of PVDF-Sn@Zn gradient electrode and its induced Zn deposition. SEM images of **b** SSM, **c** Sn@Zn, **d** imprinted Sn@Zn, and **e**, **f** PVDF-Sn@Zn gradient electrodes. **g–i** The corresponding EDS mappings of **f**. **j–m** Cross-sectional SEM and the corresponding EDS mapping of PVDF-Sn@Zn gradient electrode. Scale bar, 30 μm for **b–e**, 60 μm for **f-i**, and 5 μm for **j–m**.

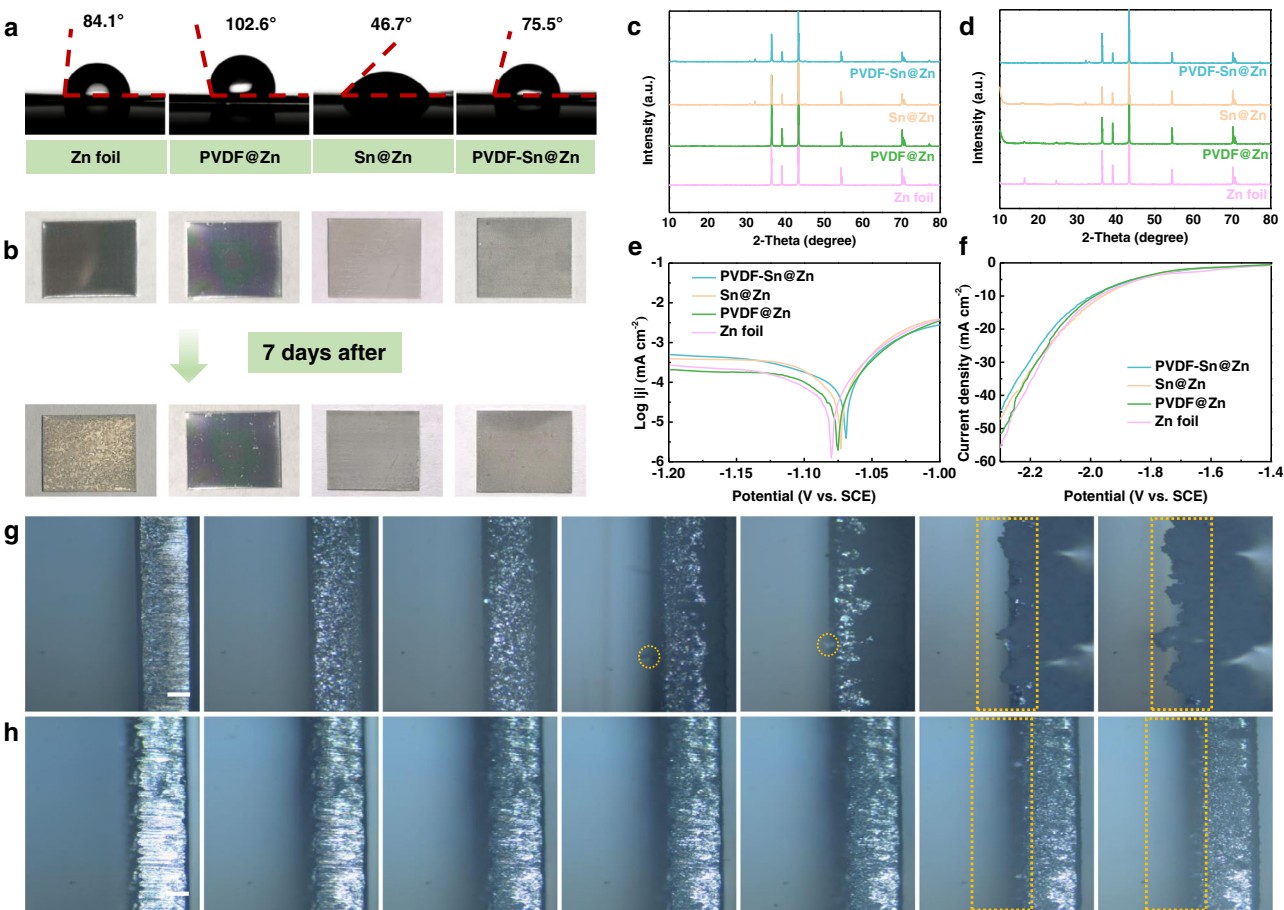

**Fig. 3 | Side reaction resistance of PVDF-Sn@Zn gradient electrode. a** Contact angles of 2 M ZnSO₄ aqueous solution on the different electrodes. **b** Digital photos of different electrodes in the initial state and after immersion in 2 M ZnSO₄ electrolyte for 7 days. XRD patterns of different electrodes **c** before and **d** after immersion in 2 M ZnSO₄ electrolyte for 7 days. **e** Corrosion and **f** HER curves of different electrodes. In situ optical observations of **g** Zn foil and **h** PVDF-Sn@Zn gradient electrodes in symmetric transparent cells at 10 mA cm⁻². Scale bar, 100 μm for **g**, **h**.

and Sn@Zn electrodes are $1.2 \times 10^{-4}$ S cm⁻¹ and 41.6 S cm⁻¹, respectively, demonstrating gradient conductivity from top to bottom in the microchannels. From the contact angle tests (Fig. 3a), PVDF@Zn exhibits a good hydrophobic property with a larger value of 102.6°, while the Sn@Zn shows a strong hydrophilic feature with a smaller value of 46.7°, which confirms a hydrophilic gradient in the PVDF-Sn@Zn electrode.

## Side reaction resistance

The side reaction of the Zn anode is closely correlated to the generation of $Zn_4SO_4(OH)_6 \cdot xH_2O$ and the occurrence of the hydrogen evolution reaction (HER) process. Figure 3b shows the digital images of the four electrodes (Zn foil, PVDF@Zn, Sn@Zn, and PVDF-Sn@Zn) before and after 7 days of immersing in 2 M ZnSO₄. The surface of the Zn foil becomes rough and the metallic luster disappears after 7 days, indicating its poor side reaction resistivity. In contrast, with the protection of PVDF and/or Sn, the other three electrodes (PVDF@Zn, Sn@Zn, and PVDF-Sn@Zn) can largely retain their initial appearance, demonstrating that both PVDF and Sn can effectively enhance the side reaction resistance. From the XRD results, the PVDF-Sn@Zn gradient electrode shows no obvious changes for the identical peaks of Sn and Zn, with the weakest new peaks after the immersing test (Fig. 3c, 3d and Supplementary Fig. 5). However, for the other three electrodes, obvious diffraction peaks at 16.1° and 25.2° appear after the immersing test, which corresponds to the (002) and (2–12) plane of the $Zn_4SO_4(OH)_6 \cdot xH_2O$ (PDF#39-0688), showing their poorer side reaction resistivity. The electrochemical test is further conducted to reveal

the side reaction resistance of the above four electrodes using 1 M Na₂SO₄ as the electrolyte. Figure 3e shows both the PVDF@Zn and Sn@Zn electrodes illustrate higher corrosion potentials than bare Zn, indicating PVDF and Sn are beneficial to inhibit Zn corrosion. The PVDF-Sn@Zn gradient electrode performs best corrosion resistivity with the highest value of −1.07 V (vs. SCE). From linear sweep voltammetry (LSV) results (Fig. 3f), the PVDF-Sn@Zn gradient electrode requires the highest overpotential of −2.21 V (vs. SCE) to reach a current density of 30 mA cm⁻², confirming its best resistivity for side reactions. The in situ optical microscopy of the Zn deposition process on Zn foil and PVDF-Sn@Zn gradient electrodes are further recorded. Due to the poor side reaction resistance and slow Zn²⁺/Zn reaction kinetics of bare Zn, the deposited Zn easily forms dendrites and significant corrosion can be observed (Fig. 3g). In comparison, the PVDF-Sn@Zn gradient electrode shows no apparent dendrite and corrosion (Fig. 3h). It is worth noting that in the PVDF-Sn@Zn gradient electrode, the pre-imprinted microchannels are gradually filled in the deposition process, forming a flat surface. The above results confirm that the PVDF-Sn@Zn electrode with gradient microchannels achieves superior side reaction resistivity.

## Zn deposition behavior

The gradient electrode also optimizes the Zn deposition behavior. Figure 4 shows the schematics (Fig. 4a) and the according SEM images (Fig. 4b–e) of the Zn deposition process in the PVDF-Sn@Zn gradient electrode. In detail, before Zn deposition, the microchannels can be clearly observed with a depth of ~14 μm (Fig. 4b). After deposition of

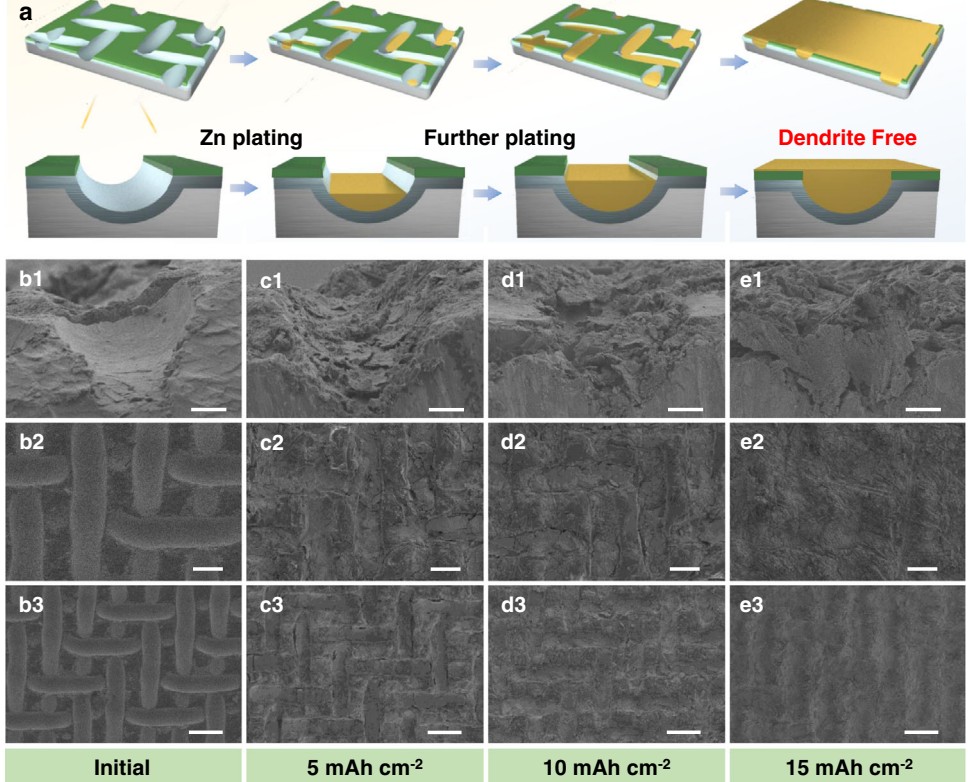

**Fig. 4 | Zn deposition morphology of PVDF-Sn@Zn gradient electrode. a** Models of PVDF-Sn@Zn gradient electrode with different deposition capacities. SEM images of PVDF-Sn@Zn gradient electrode after Zn deposition with capacities of **b1–b3** 0, **c1–c3** 5, **d1–d3** 10, and **e1–e3** 15 mAh cm⁻², respectively. Scale bar, 10 μm for **b1–e1**, 30 μm for **b2–e2**, and 60 μm for **b3–e3**.

Zn with a capacity of 5 mAh cm⁻², most of the Zn is deposited at the bottom of the microchannels and no obvious deposition emerges on the top PVDF layer (Fig. 4c), indicating the gradient design preferentially induces Zn to nucleate and grow at the bottom of the microchannels. As the deposition capacity increases, the microchannels are gradually filled (Fig. 4d) and a uniform and flat surface is observed when the deposition capacity reaches 15 mAh cm⁻² (Fig. 4e). Notably, no uneven protuberance is observed. The desirable bottom-up deposition behavior of the PVDF-Sn@Zn gradient electrode can effectively reduce the risk of short circuits caused by the dendrite piercing the separator, thus promoting stable cycling at high current density/capacity. As a further comparison, the Zn deposition behaviors on the other three electrodes are also studied. As shown in Supplementary Fig. 6, the surface of the Zn foil becomes uneven accompanied by apparent dendrites forms even at a low deposition capacity of 5 mAh cm⁻². Benefiting from the good inertness of PVDF, the PVDF@Zn electrode achieves certain improvements but still shows significant inhomogeneous deposition (Supplementary Fig. 7). Since Sn has good zincophilicity and high redox potential, the Sn@Zn electrode keeps a uniform surface at a deposition capacity of 5 mAh cm⁻². However, apparent cracks are observed with higher deposition capacities (Supplementary Fig. 8).

**Electrochemical performance**
Asymmetric cells (noted as Cu//Zn, PVDF@Cu//Zn, Sn@Cu//Zn, and PVDF-Sn@Cu//Zn) and symmetric cells (noted as Zn foil//Zn foil, PVDF@Zn//PVDF@Zn, Sn@Zn//Sn@Zn, and PVDF-Sn@Zn//PVDF-Sn@Zn) are further assembled and tested. As shown in Fig. 5a, b, the PVDF@Cu electrode exhibits a much higher nucleation overpotential (45.9 mV) than that of the Sn@Cu electrode (21.1 mV), due to the lower conductivity and worse hydrophilicity of PVDF compared with Sn.

Notably, the PVDF-Sn@Cu gradient electrode exhibits the smallest nucleation overpotential (18.9 mV), confirming the optimized gradient structure is beneficial for Zn nucleation and deposition. From the electrochemical impedance spectroscopy (EIS) plots of the symmetric cells in Fig. 5c, the Sn@Zn//Sn@Zn cell exhibits a significantly smaller electron transfer resistance than that for the PVDF@Zn//PVDF@Zn cell. For the PVDF-Sn@Zn//PVDF-Sn@Zn cell, it shows a similar value to that of the Sn@Zn//Sn@Zn cell, indicating that the top surface of PVDF will not deteriorate the fast electron transfer in the PVDF-Sn@Zn gradient electrode. The chronoamperometry (i-t) curves also show the PVDF-Sn@Zn gradient electrode possesses superior 3D diffusion property, which is beneficial for the uniformity and selectivity of Zn deposition (Supplementary Fig. 9). From Supplementary Fig. 10, the PVDF-Sn@Cu//Zn cell maintains stable CE of over 99% for more than 500 cycles, indicating the gradient microchannel design also results in high Zn plating/stripping reversibility.

Benefiting from the promising side reaction resistivity and desirable bottom-up Zn deposition behavior, the PVDF-Sn@Zn gradient electrode is anticipated to illustrate good cycling stability. As shown in Fig. 5d, PVDF-Sn@Zn//PVDF-Sn@Zn cell can maintain stable small voltage hysteresis (less than 15 mV for the first 600 cycles) for over 1200 h at current density/capacity of 1 mA cm⁻²/1 mAh cm⁻². On the contrary, the cells of Zn foil//Zn foil, PVDF@Zn//PVDF@Zn, and Sn@Zn//Sn@Zn experience short-circuit after only 92, 294, and 445 h, respectively (the dendrites pierce the separator and cause battery failure). The voltage hysteresis exhibited by the PVDF-Sn@Zn//PVDF-Sn@Zn cell is superior to that of the recently reported Zn anodes (Supplementary Table 1). The PVDF-Sn@Zn gradient electrode also demonstrates good cycling stability at high current densities of 2 and 5 mA cm⁻², as shown in Supplementary Figs. 11, 12. It is worth

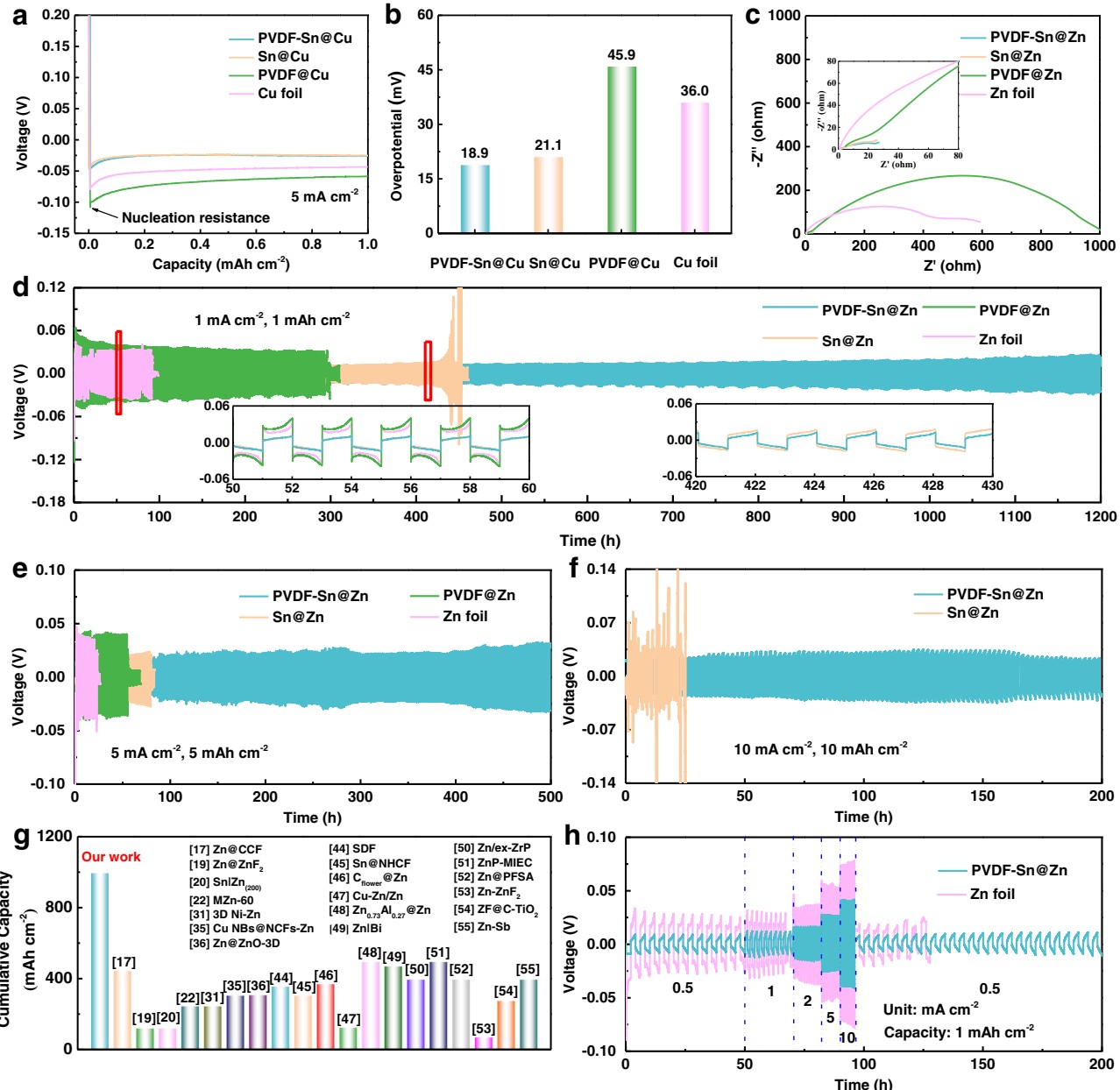

**Fig. 5 | The electrochemical performance of different electrodes. a** Voltage–time curves for Zn deposition at 5 mA cm⁻² on different electrodes. **b** The summary of Zn nucleation overpotentials for different electrodes. **c** EIS plots of the symmetrical cells assembled by different electrodes. Voltage profiles of Zn//Zn symmetrical cells at current densities/capacities of **d** 1 mA cm⁻²/1 mAh cm⁻², **e** 5 mA cm⁻²/5 mAh cm⁻², and **f** 10 mA cm⁻²/10 mAh cm⁻². **g** Comparison of the PVDF-Sn@Zn gradient electrode with recently reported Zn anodes using artificial interfacial layer strategy or 3D structured design. **h** Rate performance of Zn foil and PVDF-Sn@Zn gradient electrode.

noting that the special bottom-up deposition behavior achieved by the gradient design allows for long-term cycling at large capacities. As shown in Fig. 5e and Supplementary Fig. 13, PVDF-Sn@Zn//PVDF-Sn@Zn cell maintains stable cycling performance of more than 500 h at a high current density/capacity of 5 mA cm⁻²/5 mAh cm⁻², which is much better than the other controlled cells. The voltage hysteresis of PVDF-Sn@Zn//PVDF-Sn@Zn cell becomes smaller at 300–400 h (from 27 to 23 mV), which can be related to changes in the electrode/electrolyte interface and the temperature of the test environment (Supplementary Fig. 14). At higher current density/capacity of 10 mA cm⁻²/10 mAh cm⁻², the PVDF-Sn@Zn//PVDF-Sn@Zn cell still stably operates for over 200 h (Fig. 5f). Notably, the current density and capacity for the stable cycled PVDF-Sn@Zn//PVDF-Sn@Zn cell are much higher

than previously reported values from Zn electrodes based on artificial interfacial layer strategy and 3D structured design (typically less than 5 mA cm⁻² and 5 mAh cm⁻², Supplementary Table 2), and the cumulative capacity (1000 mAh cm⁻² at 10 mA cm⁻²/10 mAh cm⁻²) obtained for the PVDF-Sn@Zn gradient electrode is also much higher than those of the recently reported Zn anodes (Fig. 5g)[17,19,20,22,31,35,36,44–55]. Besides, the PVDF-Sn@Zn//PVDF-Sn@Zn cell also possesses stable performance with a wide range of 0.5 to 10 mA cm⁻² (Fig. 5h), indicating its good rate capability. The better cycling performance of the PVDF-Sn@Zn gradient electrode than the other three controlled samples can be attributed to the optimization of the current density and Zn²⁺ ion concentration distribution through the gradient design, as shown in multiphysics simulations (Supplementary Fig. 15). The non-

conductive PVDF layer on the top of the PVDF-Sn@Zn gradient electrode can effectively reduce the current density on the top surface of the electrode, thereby inhibiting the top growth of dendrites. In addition, the 3D gradient structure can not only accelerate the $Zn^{2+}$ ions diffusion and increase the $Zn^{2+}$ ion concentration on the electrode surface, but also induces the $Zn^{2+}$ ions to incline to microchannels, resulting in uniform deposition of Zn in the microchannels. To achieve good stability, the cycling performance of the PVDF-Sn@Zn gradient electrodes prepared with different mesh sizes of the SSM are measured (Supplementary Fig. 16). As shown in Supplementary Fig. 17, benefiting from the favorable porosity, the PVDF-Sn@Zn gradient electrode obtained with 500-mesh SSM shows a desirable cycling performance. The thicknesses of the PVDF and Sn layers are also optimized. Supplementary Fig. 18 and 19 show that a PVDF thickness of 2.4 µm and an Sn layer thickness of 4 µm are favorable.

The morphologies of the PVDF-Sn@Zn gradient electrode after cycling have been further investigated. It can be found that the PVDF-Sn@Zn gradient electrode maintains good Zn plating/stripping reversibility during repeated charge-discharge, and no obvious dendrite is observed after 500 cycles (Supplementary Fig. 20). In contrast, the Zn foil shows obvious dendrites after only 50 cycles (Supplementary Fig. 21), while PVDF@Zn and Sn@Zn also show hazardous dendrite after 200 and 400 cycles, respectively (Supplementary Figs. 22, 23).

### Full battery performance

To investigate the practical application of the gradient electrode, full Zn-ion batteries are assembled by using $MnO_2$@C (in situ growth of $MnO_2$ on vertical graphene-decorated carbon cloth) cathode coupled with the PVDF-Sn@Zn gradient anode (noted as $MnO_2$@C//PVDF-Sn@Zn), as shown in Fig. 6a. The XRD result confirms the successful

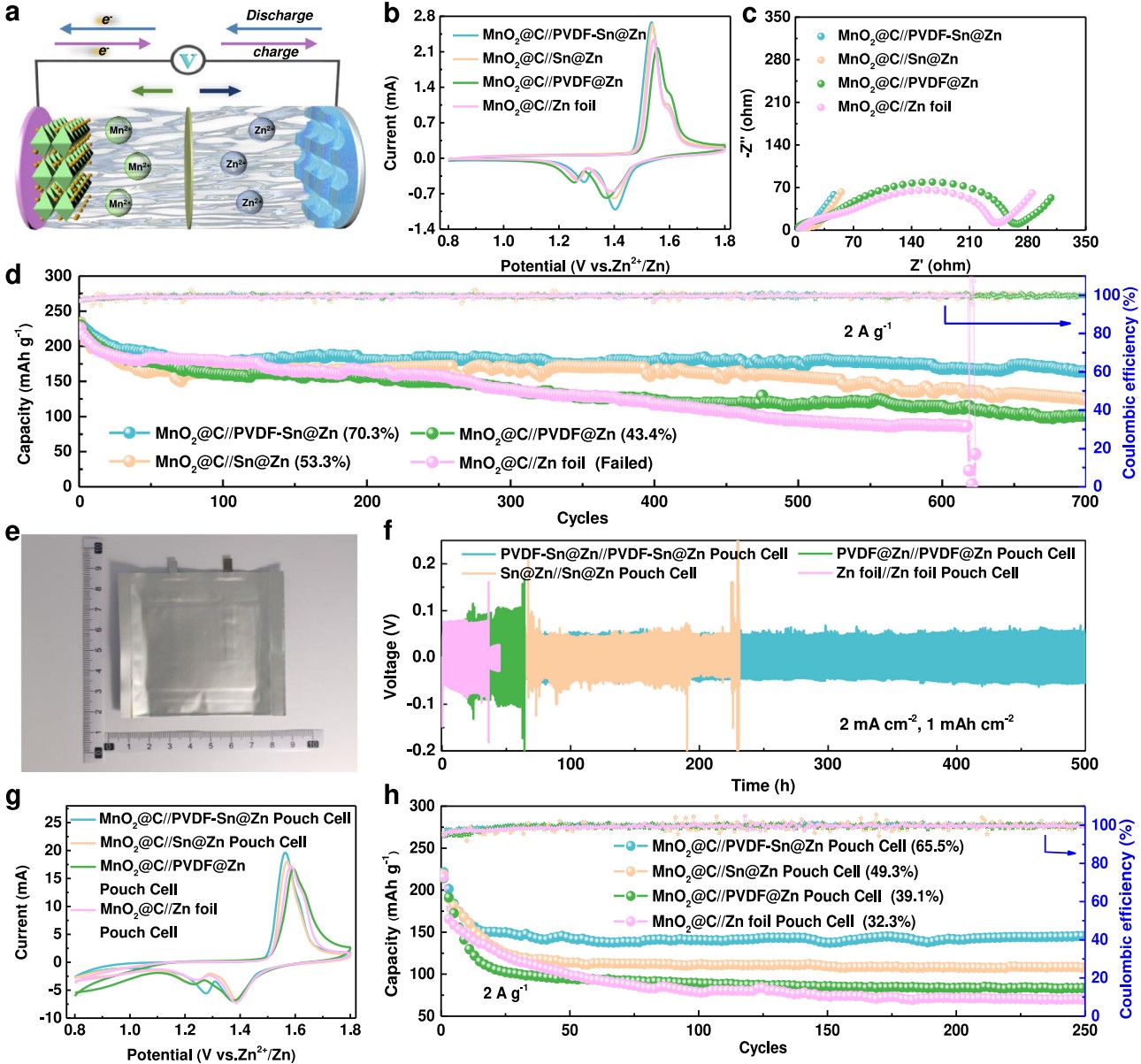

**Fig. 6 | Electrochemical performance of full cells and pouch cells. a** Schematic of the Zn-ion battery device. **b** CV curves during the second cycle, **c** EIS plots, and **d** long-term cycling stability at a specific current 2 A g⁻¹ of different full cells. **e** Photo of $MnO_2$@C//PVDF-Sn@Zn pouch cell. **f** Stability of symmetric pouch cells at a current density of 2 mA cm⁻² (40 mA) with a capacity of 1 mAh cm⁻² (20 mAh). **g** CV curves and **h** cycling stability of full pouch cells at a specific current of 2 A g⁻¹.

synthesis of $MnO_2$ (Supplementary Fig. 24), and the $MnO_2$@C electrode exhibits a porous structure, which is conducive to rapid ion/electron transport (Supplementary Fig. 25). From the cyclic voltammetry (CV) curves in Fig. 6b, $MnO_2$@C//PVDF-Sn@Zn cell shows obviously smaller voltage polarization than the other three cells, demonstrating its favorable $Zn^{2+}$/Zn reaction kinetics. The EIS results also confirm that the $MnO_2$@C//PVDF-Sn@Zn cell has the smallest charge transfer resistance (Fig. 6c). The small polarization and charge transfer resistance can be attributed to the inhibition of side reactions and the regulation of Zn deposition behavior by the gradient microchannel design. The cyclic performance of the four cells is shown in Fig. 6d. $MnO_2$@C//PVDF-Sn@Zn cell can retain 70.3% of its initial capacity in the 700th cycles, which is much better than the other three cells ($MnO_2$@C//Sn@Zn-53.3%, $MnO_2$@C//PVDF@Zn-43.4% and $MnO_2$@C//Zn foil-failed). The $MnO_2$@C//PVDF-Sn@Zn cell also exhibits good rate performance at a wide range of 1 to $10\,A\,g^{-1}$, where its capacities are consistently the highest among the four cells (Supplementary Fig. 26). The good cycling stability and rate performance of the $MnO_2$@C//PVDF-Sn@Zn cell again confirm the merits of the gradient design.

Benefiting from the simplicity, low cost, and scalability of the imprinted gradient design, the PVDF-Sn@Zn gradient electrode can be easily scaled-up. To further evaluate the practicability of the PVDF-Sn@Zn gradient electrode, pouch cells with large size of $4\,cm^2 \times 5\,cm^2$ are further assembled and their electrochemical performance are tested under pressure (Fig. 6e and Supplementary Figs. 27, 28). Noting that the size of the electrode is enlarged 20 times compared with the coin cell, thus the problems from the side reactions and dendrite growth shall be more serious. As shown in Fig. 6f, the PVDF-Sn@Zn//PVDF-Sn@Zn pouch cell maintains a small voltage hysteresis and stable cycling for more than 500 h at a current density/capacity of $2\,mA\,cm^{-2}$ (40 mA)/ $1\,mAh\,cm^{-2}$ (20 mAh), better than Sn@Zn//Sn@Zn (231 h), PVDF@Zn//PVDF@Zn (63 h), and Zn foil//Zn foil (36 h) pouch cells. $MnO_2$@C//PVDF-Sn@Zn pouch cell also exhibits superior reaction kinetics with lower voltage polarization than that for the other three pouch cells (Fig. 6g). The stability test reveals that the $MnO_2$@C//PVDF-Sn@Zn pouch cell can retain more than 65.5% of the initial capacity in the 250th cycle, which is also much better than the values for $MnO_2$@C//Sn@Zn (49.3%), $MnO_2$@C//PVDF@Zn (39.1%), and $MnO_2$@C//Zn foil (32.3%) pouch cells (Fig. 6h). From the morphologies after cycling, the large-size PVDF-Sn@Zn gradient electrode possesses promising Zn plating/striping reversibility, as a flat and smooth surface can be maintained, while the surface of the Zn foil becomes significantly rougher with dendrites (Supplementary Fig. 29). The $MnO_2$@C//PVDF-Sn@Zn pouch cell can also exhibit a high energy density of $163.3\,Wh\,kg^{-1}$ at a power density of $2286.1\,W\,kg^{-1}$. In addition, the assembled $MnO_2$@C//PVDF-Sn@Zn pouch cell possesses good mechanical stability, which can power 16 LEDs and maintain the brightness in bended state (Supplementary Fig. 30).

## Discussion

In summary, we reported an imprinted gradient Zn anode that well integrates conductive and hydrophilic gradients and simultaneously regulates the side reactions and Zn deposition behavior. The top hydrophobic PVDF layer and the bottom stable Sn layer synergistically enhanced the corrosion resistance of the Zn anode and inhibited the HER process. The gradient microchannel design also effectively optimized the electric field distribution, $Zn^{2+}$ ion flux, and local current density, thus achieving desired bottom-up deposition behavior for Zn metal and effectively avoiding the problem of top dendrite growth. As a result, the PVDF-Sn@Zn gradient electrode maintained stable cycling for more than 200 h at a high current density/capacity of $10\,mA\,cm^{-2}$/ $10\,mA\,cm^{-2}$, outperforming previously reported Zn anodes with an

artificial protecting layer or 3D structure. Our imprinting gradient design can be easily scaled-up, which provides a promising way for dendrite-free metal batteries at high current densities and high capacities.

## Methods

### Preparation of Sn@Zn electrode
The Sn@Zn electrode was fabricated by a simple displacement reaction. First, a pre-cleaned Zn foil (200 μm) was covered with an insulating Kapton film to ensure that the reaction can only take place on one side. The treated Zn foil was then placed in a 0.4 M $SnCl_4$ solution and reacted for 5 min. Finally, the Sn@Zn electrode was obtained after drying.

### Preparation of PVDF@Zn electrode
The PVDF@Zn electrode was prepared by spin coating. First, 200 mg of PVDF was added to 2 mL of N-methyl−2-pyrrolidine (NMP) to form a homogeneous solution. Then, the solution was spin-coated (4000 rpm, 1 min) onto Zn foils to form a uniform coating layer. Finally, the PVDF@Zn electrode was obtained after evaporation of the NMP solvent.

### Preparation of PVDF-Sn@Zn gradient electrode
The PVDF-Sn@Zn gradient electrode was fabricated through the imprinting and coating process. First, the obtained Sn@Zn electrode and the pre-cleaned stainless-steel mesh (SSM, 500-mesh) were passed through a roller press and imprinted together. Then, a layer of PVDF was casted on the surface of the imprinted Sn@Zn and dried. Finally, the PVDF-Sn@Zn gradient electrode was obtained after removing the SSM. Noting the SSM can be repeatedly used.

### Preparation of Sn@Cu electrode
The preparation of the Sn@Cu electrode was similar to that for the Sn@Zn electrode, except for an extra pre-Zn deposition process. The solution for predetermining deposited Zn metal was 0.2 M $ZnSO_4$ and 0.5 M $Na_3C_6H_5O_7$·$2H_2O$. First, chronoamperometry was applied to electrodeposit Zn on Cu foil at −1.4 V (vs. saturated calomel electrode) for 15 min. Then, the same Sn chemical displacement reaction was applied to achieve the Sn@Cu electrode. Finally, the Sn@Cu electrode was obtained after drying.

### Preparation of PVDF@Cu electrode
The preparation of the PVDF@Cu electrode was similar to that of the PVDF@Zn electrode, except that the Zn is replaced with Cu.

### Preparation of PVDF-Sn@Cu gradient electrode
The preparation of the PVDF-Sn@Cu gradient electrode was similar to that of the PVDF-Sn@Zn gradient electrode, except that the Sn@Zn is replaced with Sn@Cu.

### Preparation of $MnO_2$@C electrode
$MnO_2$@C electrode was prepared by hydrothermal method. In detail, 67 mg of $KMnO_4$ with 0.5 mL HCl were dissolved in 60 mL of distilled water and transferred into a 100 mL Teflon-lined stainless autoclave, with a piece of vertical graphene-decorated carbon cloth (VG@CC) immersed into the solution. After maintaining at 85 °C for 20 min, the $MnO_2$@C electrode was obtained (the loading mass was about $0.8\,mg\,cm^{-2}$).

### Materials characterization
XRD studies were conducted using Bruker D8 Advanced with radiation from a Cu target. Field emission scanning electron microscopy (FE-SEM) studies were conducted using FEI Verios G4 (20 kV). XPS spectra were obtained from Kratos (Axis Supra). Four-probe resistivity results were obtained from tester H7756 and contact angle results were

obtained from DSA 25 (KRUSS). An optical microscope equipped with EMCCD and a monochromator was used for the in situ optical observations.

## Electric field simulation

The proportional schematics of electric field distribution, $Zn^{2+}$ ion flux, and local current density during cycling were studied and compared by establishing a 2D model. The ionic conductivity of 2 M $ZnSO_4$ electrolyte was set to be about $3\,S\,m^{-1}$ as reported. According to the Butler-Volmer equation (Eq. 1), the local current density can be calculated as follows:

$$i_{loc} = i_0(\exp(\frac{\alpha_a F\eta}{RT}) - exp(\frac{-\alpha_c F\eta}{RT})) \tag{1}$$

Where $i_{loc}$ is the current density of the electrode, $i_0$ represents the exchange current density, $\alpha_a$ represents the charge transfer coefficient in the anode direction, $F$ is the Faraday constant, $\eta$ is the activation overpotential, $R$ represents the ideal gas constant, $T$ is the temperature in Kelvin, and $\alpha_c$ represents the charge transfer coefficient in the cathode direction.

The diffusion and electric field migration equations were used to study the ion migration:

$$N_{Zn} = -D_{Zn}\nabla c_{Zn} - Z_{Zn}\mu_{m,Zn}Fc_{Zn}\nabla\phi_l \tag{2}$$

where $N_{Zn}$ is the $Zn^{2+}$ diffusion flux, $D_{Zn}$ represents the $Zn^{2+}$ diffusion coefficient, $\nabla c_{Zn}$ is the $Zn^{2+}$ concentration gradient. $\mu_{m,Zn}$ represents the Mobility of $Zn^{2+}$, $Z_{Zn}$ is the $Zn^{2+}$ band charge and $\phi_l$ represents the electric potential in solution.

## Electrochemical measurement

All electrochemical tests were performed at room temperature. For electrochemical tests, CR-2032-type coin cells were assembled using a glass fiber filter (GF-A, Whatman) as the separator. Symmetrical cells using 2 M $ZnSO_4$ electrolyte were used to study the Zn deposition behavior, and a current density of 10 mA $cm^{-2}$ was set for the Zn deposition. To study the nucleation process and Coulombic efficiency, asymmetrical cells were assembled using PVDF-Sn@Cu as the cathode, Zn foil as the anode, and 2 M $ZnSO_4$ as the electrolyte, and the stripping cutoff voltage was set at 0.5 V (vs. $Zn^{2+}$/Zn). In the full-cell Zn-ion battery tests, 2 M $ZnSO_4$ with 0.1 M $MnSO_4$ were used as the electrolyte. The capacities of the full cells were obtained based on the weight of $MnO_2$ active materials. All the controlled samples, including Zn foil, PVDF@Zn, and Sn@Zn electrodes, were investigated in the same condition.

The energy density and power density of $MnO_2@C//PVDF-Sn@Zn$ pouch cell are calculated by the following equations:

$$E = \frac{\int IUdt}{m} \tag{3}$$

$$P = \frac{E}{\triangle t} \tag{4}$$

where E is the energy density, P represents the power density, I is the discharging current, dt represents the time differential, $\triangle t$ is the discharging time, and m refers to the mass loading of $MnO_2$.

## Data availability

The data that support the findings of this study are available from the corresponding author upon reasonable request.

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

## Acknowledgements

The authors acknowledge the financial support of the National Key Research and Development Program (2022YFE0121000), Fundamental Research Funds for the Central Universities, and the Innovation Foundation for Doctor Dissertation of Northwestern Polytechnical University (no. CX2021042).

## Author contributions

C.G. conceived the idea. Q.C. and Y.G. carried out the materials synthesis and electrochemical characterization. Y.G. carried out theoretical simulations. J.P., X.Z., Y.W., and J.C. provided important experimental insights. All the authors discussed the results and contributed to writing the manuscript.

## Competing interests

The authors declare no competing interests.
