## [Peer Review File · Nature Communications]

Gradient design of imprinted anode for stable Zn-ion batteriesREVIEWER COMMENTS

Reviewer #1 (Remarks to the Author):

The popularization of rechargeable zinc-ion batteries is seriously hindered by the zinc dendrite growth, electrolyte corrosion and side reactions on the Zn anode side. In this manuscript, the authors reported a simple imprinting method to construct imprinted gradient Zn anode (PVDF-Sn@Zn) that integrate gradient conductivity and hydrophilicity for long-term stable Zn-ion batteries. Systematic and comprehensive electrochemical and structural characterizations were applied to reveal the protection mechanism of double-gradient structure PVDF-Sn on Zn anode. Moreover, large-area (4*5 cm²) MnO₂@C//PVDF-Sn@Zn pouch cell assembled based on on PVDF-Sn@Zn anode exhibits excellent performance indicators. Overall, this work proposed a novel and practical method to construct long-term stable Zn anode, but there are many defects and doubts in the manuscript that need to be revised. Therefore, I recommend this manuscript can be reconsidered after major revision.

Comments:

1. PVDF (such as Adv. Funct. Mater. 2021, 31, 2001867; Energy Storage Mater. 2022, 47, 602–610; Nano Energy 2022, 103, 107805) and Sn (such as Sci. Adv. 2022, 8, eabm5766; Chem. Eng. J. 2022, 437, 135246; Energy Storage Mater. 2022, 51, 259-265; Appl. Phys. Rev. 2022, 9, 011401) in the double-gradient protective layer have been independently studied as protective layers recently. So, it is necessary for the authors to state the novelty of this work.
2. Zincophilicity and zincophobicity are important indicators for the construction of Zn anode protective layer, but they are not covered in this work.
3. The authors claimed that the PVDF-Sn@Zn electrode exhibits good flatness, but many obvious wrinkles can be observed from its optical photo (Supplementary Figure 1).
4. The authors claimed that no additional peak can be detected in the XRD pattern of the PVDF-Sn@Zn gradient electrode after the immersing test. However, it is obvious that the XRD pattern of PVDF-Sn@Zn gradient electrode has several additional peaks at 15-30°.
5. The authors gave the Zn deposition model of PVDF-SN@Zn gradient electrode in Figure 4, but did not prove its rationality.
6. The current density was not indicated in Supplementary Fig. 17.
7. Energy density is an important performance indicator of the pouch cell, and it is necessary for the authors to give.

Reviewer #2 (Remarks to the Author):

In this manuscript the authors describe a four step series of chemical and mechanical processes for modifying a conventional Zn foil anode to address known challenges with this cell chemistry (poor plating/stripping reversibility, dendrite growth, cycling stability, utilization, and corrosion/side reactions).

This series consists of the following steps:

- 1.) A simple displacement reaction is applied to introduce Sn to the surface of Zn foil
- 2.) The Sn@Zn foil is then imprinted with a stainless steel mesh to produce a microchannel surface pattern
- 3.) A layer of PVDF is cast on the surface of the Sn@Zn and then dried
- 4.) The stainless steel mesh is removed from the surface

The authors propose that this combined approach results in a structure with intrinsic gradients in conductivity and hydrophilicity which are key to observed performance improvements. In addition to characterization of the fully modified anode (PVDF-Sn@Zn), the authors screen the contributions of each of these modifications (Sn@Zn, PVDF@Zn, imprinted Zn) to the overall performance with a variety of structural/electrochemical characterization approaches (SEM, EDS, contact angle, XRD, Corrosion/HER curves, and symmetric (Zn vs. Zn)/half (Zn vs. Cu)/full (Zn vs. MnO₂/C) cell studies). Finite element simulations are also included to show impact on electric field distribution, ion flux, and current density.

Regarding novelty, the author's correctly mention in the introduction that 3D Zn anode designs and Sn-based current collector or anode coatings for Zn have been explored in other studies, and there are a few other papers recently published in the literature on both of these topics. The same can be said for PVDF coatings on Zn, such as the following report: (<https://www.sciencedirect.com/science/article/pii/S1385894721001832>). Thus, while the individual approaches are not novel, it is the combination of these approaches that is the unique feature of this manuscript and would be of some interest to the Zn community.

In general, the work supports the conclusions, the methodology is sound, and there is enough detail provided in the methods for the work to be reproduced. This paper is suitable for publication in Nature Communications if the following questions and/or minor revisions can be addressed:

- 1.) Are the anode images in Figure S11 shown in the charged (plated) or discharged (stripped) state? This should be included in the caption for context.
- 2.) Are the pouch cells cycled under any externally applied pressure? Was the impact of stack pressure explored?
- 3.) Have the authors explored any other mesh sizes for the surface modifications? How was 500 mesh selected?
- 4.) Is the Sn surface treatment self-limiting in terms of thickness or was there some optimization for this step in the process?
- 5.) Assuming the PVDF-coated areas of the PVDF-Sn@Zn imprinted Zn anode are electrochemically inactive, do the authors have an estimate for the enhancement/reduction of the electrochemical surface area as compared to the geometric area of the normal foil? As the authors described in the introduction, surface area enhancement is one approach for reducing current density, but it is not clear if the SS mesh imprinting strategy leveraged in this work has a net enhancement or reduction in electrochemically active surface area compared to the baseline bare foil or imprinted/PVDF coated foil. Some discussion and basic calculations here would be helpful for interpreting the results.
- 6.) What are the rates shown in Figure S17? Please make sure these are appropriately normalized and the relevant mass (either electrode level or active material level) are given.
- 7.) What is the expected phase of the MnO₂ cathode used in the full cell demonstration based on the XRD in Figure S15?
- 8.) The authors perform a corrosion analysis in Figure 3e. It would be useful to extract the associated exchange current densities using a Tafel analysis. This would also be interesting to compare to the values used for the finite element simulations.
- 9.) Do the authors have any electrochemical data comparing the imprinted Zn foil vs. regular Zn foil? At the beginning of the results section under "Gradient Electrode Design" the authors mention comparisons between the bare/3D Zn but only support these conclusions with modeling results (Fig. 1a/1b).

Reviewer #3 (Remarks to the Author):

In this manuscript, Guan et al. manuscript reported an imprinted gradient Zn anode (noted as PVDF-Sn@Zn), where the PVDF-Sn@Zn gradient electrode showed a cycling of 1200 h at low current density/capacity of 1 mA cm⁻²/1 mAh cm⁻² and 200 h at high current density/capacity of 10 mA cm⁻²/10 mAh cm⁻². However, when coupled with the MnO₂@C electrode as a full cell, the cycling performance of the PVDF-Sn@Zn was only slightly better compared with that of Sn@Zn. Moreover, the cycling stability of Sn@Zn||Sn@Zn pouch cell, PVDF@Zn ||PVDF@Zn pouch cell, MnO₂@C||Sn@Zn pouch cell and MnO₂@C|| PVDF@Zn pouch cell had not been provided. I cannot recommend this work for publication in Nat. Commun. at this stage. Here are my specific points:

1. In Line 143-144, the authors stated that "the thickness of the upper PVDF layer is about 2.4 μm and the Sn layer is about 4 μm ". It is unclear that if these reported values of thickness were optimized parameters for the performance or not. Why did the authors set up the thickness of PVDF layer and Sn layer to be these values? I would suggest the authors to provide additional experimental results to optimize the setup.

2. In Line 215-216, the authors stated that "in Fig. 5d, PVDF-Sn@Zn||PVDF-Sn@Zn cell can maintain stable small voltage hysteresis (less than 15 mV) for over 1200 h ..." However, the voltage hysteresis for PVDF-Sn@Zn in Fig. 5d varied significantly with the testing time. Why was that? In addition, I would suggest the authors to compare the voltage hysteresis after long cycling tests of this work with that of other works reported previously.

3. In Fig. 5e, the voltage hysteresis for PVDF-Sn@Zn turned smaller at 300-400 h compared with that at 0-300 h (there was a similar phenomenon in Fig. 5f). What were the possible reasons? Please discuss it.

4. In Fig. 5f, the whole testing duration was 200 h; nevertheless, the voltage hysteresis for PVDF-Sn@Zn only turned stable after 160 h. I would suggest the authors to extend the testing hours at 10 mA cm^{-2} . Moreover, how about the voltage profiles of PVDF-Sn@Zn||PVDF-Sn@Zn cell at larger current densities/capacities?

5. In Fig. 6d, it seems that the capacity for PVDF-Sn@Zn was just slightly higher than that for Sn@Zn. Does this mean that the role of PVDF in PVDF-Sn@Zn is minor? Moreover, the long-term cycling stability after 630 cycles was missing. However, the authors stated in Line 250-251 "... initial capacity after 630 cycles, which is much better than the other three cells". This is less convincing.

6. In Fig. 6f-h, please provide the corresponding results of PVDF@Zn and Sn@Zn.

7. It would be helpful if the authors could provide theoretical calculations comparing PVDF-Sn@Zn with Zn foil, PVDF@Zn and Sn@Zn to validate the advantages of the PVDF-Sn@Zn gradient anode.

Response to reviewer's comments:

We thank the reviewers for the thorough evaluation of our manuscript. The valuable comments have enabled us to further improve the overall quality of the manuscript. The revisions are marked in red font color in the revised manuscript, and the point-to-point responses to the reviewers' comments are as follows.

Reviewer 1

The popularization of rechargeable zinc-ion batteries is seriously hindered by the zinc dendrite growth, electrolyte corrosion and side reactions on the Zn anode side. In this manuscript, the authors reported a simple imprinting method to construct imprinted gradient Zn anode (PVDF-Sn@Zn) that integrate gradient conductivity and hydrophilicity for long-term stable Zn-ion batteries. Systematic and comprehensive electrochemical and structural characterizations were applied to reveal the protection mechanism of double-gradient structure PVDF-Sn on Zn anode. Moreover, large-area (4*5 cm²) MnO₂@C//PVDF-Sn@Zn pouch cell assembled based on PVDF-Sn@Zn anode exhibits excellent performance indicators. Overall, this work proposed a novel and practical method to construct long-term stable Zn anode, but there are many defects and doubts in the manuscript that need to be revised. Therefore, I recommend this manuscript can be reconsidered after major revision.

____ Thank you very much for the positive support and valuable comments. The manuscript has been carefully revised according to your suggestions. The detailed responses are as follows.

Comments 1:

PVDF (such as *Adv. Funct. Mater.* 2021, 31, 2001867; *Energy Storage Mater.* 2022, 47, 602-610; *Nano Energy* 2022, 103, 107805) and Sn (such as *Sci. Adv.* 2022, 8, eabm5766; *Chem. Eng. J.* 2022, 437, 135246; *Energy Storage Mater.* 2022, 51, 259-265; *Appl. Phys. Rev.* 2022, 9, 011401) in the double-gradient protective layer have been independently studied as protective layers recently. So, it is necessary for the authors to state the novelty of this work.

____ We thank the reviewer for the valuable comment. Yes, the preparation of an artificial protective layer is an effective method to protect Zn metal from electrolyte corrosion and optimize Zn deposition behavior, thereby prolonging the cycle life of Zn anode. All the works mentioned by the reviewer belong to this strategy, and PVDF (Sn) is reported efficient. However, such reported planar artificial protective layers normally only worked effectively at low current densities/capacities (commonly less

than 5 mA (h) cm⁻²). At high current densities/capacities, the rapid growth of dendrites (hot spot effect) can be observed where the local electric field becomes large and protrusions appear on the electrode surface. Therefore, rational electrode design that reduces the electric field intensity on the electrode surface and optimizes the Zn²⁺ flux is urgently needed to improve the stability of Zn anode. In addition, the growth of Zn dendrites is thermodynamically favorable, thus reducing the impact of dendrite growth on the cycle performance of Zn electrodes is also an important factor to be considered. With the above considerations, in this work, different from previously reported works, we report a facilely imprinted gradient Zn anode that well integrates gradient conductivity and hydrophilicity. The hydrophobic PVDF layer effectively prevents Zn metal from corrosion in the electrolyte, while the Sn layer with a high redox potential (Sn²⁺/Sn, -0.136 V vs SHE) inhibits the side reactions of Zn. More importantly, the gradient conductivity effectively induces electric field distribution, Zn²⁺ ion flux and local current density toward the bottom of the microchannels, thus achieving desired bottom-up deposition behavior for Zn metal. Therefore, not only controllable and uniform Zn deposition is achieved, but also the possible short circuit from top dendrite growth is prevented. As a result, PVDF-Sn@Zn gradient electrode shows stable cycling of 1200 h at low current density/capacity of 1 mA cm⁻²/1 mAh cm⁻² and 200 h at high current density/capacity of 10 mA cm⁻²/10 mAh cm⁻².

In all, although we used a common protective material of PVDF, the electrode design strategy and the mechanism of performance improvement is quite different. We have emphasized the novelty of our work in the revised manuscript.

Comments 2:

Zincophilicity and zincophobicity are important indicators for the construction of Zn anode protective layer, but they are not covered in this work.

____ We appreciate the reviewer for the valuable comment. Yes, the rational design of electrodes with a zincophilic gradient is also an effective way to improve the stability of Zn anodes, similar conclusions have been confirmed in the design of lithium anodes. For example, Pu et al.¹ achieved a long-term stable lithium anode with lithophilic gradient by electrically passivating the top of a porous nickel scaffold and chemically activating the bottom of the scaffold. In this work, it is worth noting that Sn has good zincophilicity, which can effectively optimize the Zn deposition flatness and improve the stability of the Zn anode.^{2,3} PVDF and Sn have obvious differences in both hydrophilicity and conductivity, which will greatly affect the nucleation and growth position of Zn. Since PVDF is a non-conductive material, Zn is not deposited on the surface of PVDF, but at the interface between PVDF

and Sn, where electron/ion transfer only occur at the surface of conductive Sn. Therefore, we do not emphasize the zincophilicity gradient considering that the deposition position is all at Sn layers. We have added more discussions on the zincophilicity and zincophobicity of Zn anode protective layer.

References

- 1 Pu, J. et al. Conductivity and lithiophilicity gradients guide lithium deposition to mitigate short circuits. *Nat. Commun.* **10**, 1896 (2019).
- 2 Li, S. et al. Toward Planar and Dendrite-Free Zn Electrodepositions by Regulating Sn-Crystal Textured Surface. *Adv. Mater.* **33**, 2008424 (2021).
- 3 Xiong, P. et al. Galvanically replaced artificial interfacial layer for highly reversible zinc metal anodes. *Appl. Phys. Rev.* **9**, 011401 (2022).

Comments 3:

The authors claimed that the PVDF-Sn@Zn electrode exhibits good flatness, but many obvious wrinkles can be observed from its optical photo (Supplementary Figure 1).

____ We thank the reviewer for the valuable comment. The PVDF-Sn@Zn electrode was fabricated by a three-step fabrication process, including displacement reaction, imprinting process and PVDF coating. Both SEM images (Fig. 2c-2e) and optical pictures (Fig. 3h) can prove that the PVDF-Sn@Zn electrode shows good flatness. However, when we tries on large-scale Zn foil, we used rolled Zn foil, the color differences caused by the small different Sn concentrations at different Zn surfaces would result in the obvious wrinkles in Supplementary Fig. 1. Also, the fast reaction between Sn⁴⁺ and Zn would result in slight difference of the thickness of Sn in different area. We believe such structure color-induced wrinkles can be reduced using large-scale flat Zn foil and better control of the Sn⁴⁺ reaction.

Comments 4:

The authors claimed that no additional peak can be detected in the XRD pattern of the PVDF-Sn@Zn gradient electrode after the immersing test. However, it is obvious that the XRD pattern of PVDF-Sn@Zn gradient electrode has several additional peaks at 15-30°.

____ We thank the reviewer for the valuable comment. After careful evaluation, we do observe additional peaks in the PVDF-Sn@Zn gradient electrode after the immersing test. And we are sorry

about the wrong statement. We further re-evaluated the XRD results of the different electrodes after the immersing test and zoomed in the curves at 15-30°. As shown in Fig. R1, the PVDF-Sn@Zn gradient electrode shows a weak diffraction peak of $Zn_4SO_4(OH)_6 \cdot xH_2O$ (ZSO), which proves to be effective in suppressing the formation of by-products. To better compare the corrosion resistance of the different electrodes, we compared the ratio of intensity of by-products to Zn metal ($I_{ZSO(002)}/I_{Zn(100)}$), where the values of Zn foil, PVDF@Zn, Sn@Zn and PVDF-Sn@Zn gradient electrodes are 8.2%, 2.6%, 2.7% and 1.8%, respectively (Fig. R1). The results further demonstrate the good corrosion resistance of the PVDF-Sn@Zn gradient electrode.

We have also added the discussions in Supplementary Information.

Fig. R1 XRD patterns of different electrodes after immersion in 2 M $ZnSO_4$ electrolyte for 7 days.

Comments 5:

The authors gave the Zn deposition model of PVDF-Sn@Zn gradient electrode in Figure 4, but did not prove its rationality.

____ Thank you for the review's comment. The Zn deposition model in Figure 4 is intended to demonstrate the bottom-up Zn deposition behavior in the microchannels in the PVDF-Sn@Zn gradient electrode. Such deposition behavior can be clearly revealed by the ex-situ SEM tests, as shown in Fig. 4b-4e. It can be found that at a low deposition capacity of 5 mAh cm^{-2} , the deposited Zn is almost presented at the bottom of the microchannels, while no obvious deposition emerges on the top where PVDF exists, proving that the gradient design preferentially induces Zn to nucleate and grow at the bottom of the microchannels (Fig. 4c). As the deposition capacity increases, the microchannels are

gradually filled (Fig. 4d) and eventually a uniform and flat surface is developed when the deposition capacity reached 15 mAh cm^{-2} (Fig. 4e).

Comments 6:

The current density was not indicated in Supplementary Fig. 17.

____ We thank the reviewer for the valuable comment. We apologize for neglecting the current densities in Supplementary Fig. 17. The current densities have been added, as shown in Fig. R2.

Fig. R2 Rate performance of different full cells.

Comments 7:

Energy density is an important performance indicator of the pouch cell, and it is necessary for the authors to give.

____ We thank the reviewer for the valuable comment. $\text{MnO}_2@\text{C}/\text{PVDF-Sn}@\text{Zn}$ pouch cell exhibits an energy density of 163.3 Wh kg^{-1} with a power density of 2286.1 W kg^{-1} at the current density of 2 A g^{-1} .

The energy density and power density of $\text{MnO}_2@\text{C}/\text{PVDF-Sn}@\text{Zn}$ pouch cell is calculated by the following equations:

$$E = \frac{\int IUdt}{m} \tag{1}$$

$$P = \frac{E}{\Delta t} \quad (2)$$

where E is the energy density, P is the power density, I is the discharging current, dt is the time differential, Δt is the discharging time, and m refers to the mass loading of MnO₂.

Reviewer 2

In this manuscript the authors describe a four step series of chemical and mechanical processes for modifying a conventional Zn foil anode to address known challenges with this cell chemistry (poor plating/stripping reversibility, dendrite growth, cycling stability, utilization, and corrosion/side reactions). This series consists of the following steps:

- 1.) A simple displacement reaction is applied to introduce Sn to the surface of Zn foil.
- 2.) The Sn@Zn foil is then imprinted with a stainless steel mesh to produce a microchannel surface pattern.
- 3.) A layer of PVDF is cast on the surface of the Sn@Zn and then dried.
- 4.) The stainless steel mesh is removed from the surface

The authors propose that this combined approach results in a structure with intrinsic gradients in conductivity and hydrophilicity which are key to observed performance improvements. In addition to characterization of the fully modified anode (PVDF-Sn@Zn), the authors screen the contributions of each of these modifications (Sn@Zn, PVDF@Zn, imprinted Zn) to the overall performance with a variety of structural/electrochemical characterization approaches (SEM, EDS, contact angle, XRD, Corrosion/HER curves, and symmetric (Zn vs. Zn)/half (Zn vs. Cu)/full (Zn vs. MnO₂/C) cell studies). Finite element simulations are also included to show impact on electric field distribution, ion flux, and current density.

Regarding novelty, the author's correctly mention in the introduction that 3D Zn anode designs and Sn-based current collector or anode coatings for Zn have been explored in other studies, and there are a few other papers recently published in the literature on both of these topics. The same can be said for PVDF coatings on Zn, such as the following report: (<https://www.sciencedirect.com/science/article/pii/S1385894721001832>). Thus, while the individual approaches are not novel, it is the combination of these approaches that is the unique feature of this manuscript and would be of some interest to the Zn community.

In general, the work supports the conclusions, the methodology is sound, and there is enough detail provided in the methods for the work to be reproduced. This paper is suitable for publication in Nature Communications if the following questions and/or minor revisions can be addressed:

____ Thank you very much for the positive support to our manuscript and the valuable suggestions. We have carefully revised our manuscript according to your comments, and the detailed answers are as follows:

Comments 1:

Are the anode images in Figure S11 shown in the charged (plated) or discharged (stripped) state? This should be included in the caption for context.

____ We thank the reviewer for the valuable comment. Figure S11 shows the morphology of the PVDF-Sn@Zn gradient electrode after cycling. The cyclic process starts with plating and ends with stripping, so the images in Figure S11 are in the stripped state. We have added the context in the caption.

Comments 2:

Are the pouch cells cycled under any externally applied pressure? Was the impact of stack pressure explored?

____ We thank the reviewer for the valuable suggestion. The cycling test of the pouch cells is performed under pressure (by pressing a 50 ml/1.8kg stainless steel reactor against the cell) to ensure better contact between cell components. We have also explored the cycling performance of pouch cells with/without pressure conditions. As shown in Fig. R3, pouch cell without pressure shows a rapid capacity drop, which can be attributed to poor contact between cell components and high ion/electron transfer resistance. In contrast, pouch cell under pressure exhibits good capacity retention after cycling.

Fig. R3 The cycling performance of MnO₂@C//PVDF-Sn@Zn pouch cells with/without pressure conditions.

Comments 3:

Have the authors explored any other mesh sizes for the surface modifications? How was 500 mesh selected?

___ We thank the reviewer for the valuable suggestion. The optimization of stainless-steel mesh (SSM) size is conducted by comparing the cycling performance of different imprinted gradient electrodes prepared with different meshes of SSM. Fig. R4 shows SEM images of the PVDF-Sn@Zn gradient electrode prepared with 80-mesh SSM. The cycling performance (Fig. R5) shows a sudden drop in voltage at about 171 h, indicating a short circuit in the cell. The relatively poor cycling performance of this electrode can be attributed to the smaller imprinted area and void space ratio, which limits the deposition modulation property and storage capacity of the electrode. We have also attempted to prepare PVDF-Sn@Zn gradient electrodes with 800-mesh SSM. However, during the rolling process, the 800-mesh SSM is easily peeled off from the Zn foil and cannot be maintained for the following coating procedure, therefore it could not be used to prepare gradient electrodes.

Fig. R4 SEM images of **a,b** imprinted Sn@Zn and **c,d** PVDF-Sn@Zn gradient electrodes (prepared with 80-mesh SSM). Scale bar, 100 μm for **a,c** and 50 μm for **b,d**.

Fig. R5 Voltage profiles of Zn//Zn symmetric cells at current densities/capacities of $5 \text{ mA cm}^{-2}/5 \text{ mAh cm}^{-2}$.

Comments 4:

Is the Sn surface treatment self-limiting in terms of thickness or was there some optimization for this step in the process?

____ We appreciate the reviewer's kind suggestion. The thickness of the Sn layer is optimized by choosing proper immersion time for the Zn foil in the SnCl_4 solution. As shown in Fig. R6a and b, Sn@Zn-30 s electrode with Sn thickness of $0.7 \mu\text{m}$ is prepared by placing Zn foil in SnCl_4 solution for 30 s. The cycling performance shows that the $\text{Sn@Zn-30 s//Sn@Zn-30 s}$ cell experiences a short circuit after only about 20 h, which can be attributed to the limited modulation property of the thin Sn layer. We also obtained a thicker Sn layer by soaking the Zn foil in SnCl_4 solution for 10 min, as shown in Fig. R6c and d. The Sn@Zn-10 min electrode shows a porous and non-uniform structure due to severe etching/reaction, and the $\text{Sn@Zn-10 min//Sn@Zn-10 min}$ cell experiences a short circuit after only about 16 h.

Based on the above discussion, it is favorable to achieve good cycling stability of the gradient electrode when the thickness of the Sn layer is $4 \mu\text{m}$.

Fig. R6 **a** Voltage profiles of Sn@Zn-30 s//Sn@Zn-30 s symmetric cell at current densities/capacities of 5 mA cm^{-2} / 5 mAh cm^{-2} . **b** Cross-sectional SEM image of Sn@Zn-30 s. **c** Voltage profiles of Sn@Zn-10 min//Sn@Zn-10 min symmetric cell at current densities/capacities of 5 mA cm^{-2} / 5 mAh cm^{-2} . **d** Cross-sectional SEM image of Sn@Zn-10 min. Scale bar, $2 \mu\text{m}$ for **b** and $5 \mu\text{m}$ for **d**.

Comments 5:

Assuming the PVDF-coated areas of the PVDF-Sn@Zn imprinted Zn anode are electrochemically inactive, do the authors have an estimate for the enhancement/reduction of the electrochemical surface area as compared to the geometric area of the normal foil? As the authors described in the introduction, surface area enhancement is one approach for reducing current density, but it is not clear if the SS mesh imprinting strategy leveraged in this work has a net enhancement or reduction in electrochemically active surface area compared to the baseline bare foil or imprinted/PVDF coated foil. Some discussion and basic calculations here would be helpful for interpreting the results.

____ We thank the reviewers for the valuable suggestion. The active area of the PVDF-Sn@Zn electrode is roughly 22.2% higher than that of the planar electrode, and the detailed discussions and basic calculations are as follows:

Based on the assumption that the PVDF-coated areas are inactive and the imprinted microchannels are active, we developed Model 1 (top view of the electrode), as shown in Fig. R7a. According to the stainless steel mesh (SSM) parameters and SEM images (stainless steel wire diameter of 28 μm and pore diameter of 25 μm), the percentage of PVDF coated part on the whole electrode are about 22.2% ($25^2/(25+28)^2*100\%$). Thus, the imprinted microchannels account for 77.8% (1-22.2%).

To obtain the increased area of the microchannel structure compared to the planar one, we built Model 2 (Fig. R7b, front view), where the perimeter of cross-section A is $\pi*28$ (assuming that half of each SSM wire is imprinted into the Zn metal). Therefore, the area of half a cylinder is about $\pi*28/2*1$, by fixing length as a unit one. The area of a planar electrode under the same conditions is about $28*1$. Thus, it can be obtained that the active area of the PVDF-Sn@Zn electrode in the microchannels increases by about 57.1% ($\pi/2-1$). Accordingly, the ratio of the active area of the PVDF-Sn@Zn electrode can be calculated as 122.2% ($77.8\%*1.571$), which means that the active area of the PVDF-Sn@Zn electrode is 22.2% higher than that of the planar electrode.

In addition, for electric field, it is ideal to assume that the entire electric field is concentrated in the microchannels. According to the electric field simulation, there is still a part of the electric field in the PVDF-coated areas, which can share the electric field intensity. The overall area of the electrode after imprinting is increased by 44.4% ($77.8\%*57.1\%$) compared with the planar electrode. Therefore, it is beneficial to reduce the electric field intensity of the whole electrode.

We have added the discussions in Supplementary Information.

Fig. R7 a Top (Model 1) and **b** front (Model 2) view of the imprinted electrode.

Comments 6:

What are the rates shown in Figure S17? Please make sure these are appropriately normalized and the relevant mass (either electrode level or active material level) are given.

___ We thank the reviewer for the valuable suggestion. We apologize for neglecting the current densities in Supplementary Fig. 17. The current densities have been added, ss shown in Fig. R2.

Fig. R2 Rate performance of different full cells.

Comments 7:

What is the expected phase of the MnO_2 cathode used in the full cell demonstration based on the XRD in Figure S15?

___ We thank the reviewer for the valuable suggestion. The synthesis of MnO_2 cathode in this work is referred to previous reported study with minor modification.¹⁻³ XRD pattern revealed that the obtained products are indexed to pure tetragonal δ - MnO_2 phase (JCPDS 80-1098).

References

- 1 Yu, N. et al. High-Performance Fiber-Shaped All-Solid-State Asymmetric Supercapacitors Based on Ultrathin MnO_2 Nanosheet/Carbon Fiber Cathodes for Wearable Electronics. *Adv. Energy Mater.* **6**, 1501458 (2016).
- 2 Cao, Q. et al. Structure-Enhanced Mechanically Robust Graphite Foam with Ultrahigh MnO_2 Loading for Supercapacitors. *Research* **2020**, 1-10 (2020).
- 3 Zhang, Y. et al. Anchoring MnO_2 on nitrogen-doped porous carbon nanosheets as flexible arrays cathodes for advanced rechargeable Zn- MnO_2 batteries. *Energy Storage Mater.* **29**, 52-59 (2020).

Comments 8:

The authors perform a corrosion analysis in Figure 3e. It would be useful to extract the associated exchange current densities using a Tafel analysis. This would also be interesting to compare to the values used for the finite element simulations.

____ We thank the reviewer for the valuable suggestion. Figure 3e shows the corrosion resistance of the PVDF-Sn@Zn gradient electrode, obtaining an exchange current density of $1.26 \times 10^{-4} \text{ mA cm}^{-2}$. The test was carried out in a 1 M Na_2SO_4 solution and the exchange current density represents the corrosion capability of the electrolyte on the electrodes. It is quite different from the exchange current density in the COMSOL simulations, where the value is related to the kinetics of Zn deposition and it is calculated using the following equation^{1,2}:

$$i \approx i_0 \frac{F}{RT} \frac{\eta}{2} \quad (3)$$

here, η is the total overpotential, and i_0 is the exchange current density. As shown in Fig. R8, the PVDF-Sn@Zn gradient electrode exhibits an exchange current density (for Zn deposition) of 13.1 mA cm^{-2} , which is used in the finite element simulations.

Fig. R8 Exchange current density from curves at different scan rates in the symmetric cell.

References

- 1 Xie, X. et al. Manipulating the ion-transfer kinetics and interface stability for high-performance zinc metal anodes. *Energy Environ Sci.* **13**, 503-510 (2020).
- 2 Guo, Y. et al. An Autotransferable g-C₃N₄ Li⁺-Modulating Layer toward Stable Lithium Anodes. *Adv. Mater.* **31**, e1900342 (2019).

Comments 9:

Do the authors have any electrochemical data comparing the imprinted Zn foil vs. regular Zn foil? At the beginning of the results section under "Gradient Electrode Design" the authors mention comparisons between the bare/3D Zn but only support these conclusions with modeling results (Fig. 1a/1b).

____ We thank the reviewer for the valuable suggestion. The cycling performance of Zn//Zn symmetric cells based on Zn foil and imprinted Zn foil (noted as Zn foil//Zn foil and imprinted Zn foil//imprinted Zn foil) is shown in Fig. R9. The imprinted Zn foil//imprinted Zn foil cell shows smaller voltage hysteresis and better cycling performance than the Zn foil//Zn foil cell, benefiting from the increased surface area and void spaces from the imprinting process. However, the imprinted Zn foil//imprinted Zn foil cell still shows a rapid voltage decay at about 61 h, which can be attributed to a short circuit caused by the growth of dendrites at the top of the electrode.

We have also added the discussions in Supplementary Information.

Fig. R9 Voltage profiles of Zn//Zn symmetric cells at current density/capacity of 5 mA cm⁻²/5 mAh cm⁻².

Reviewer 3

In this manuscript, Guan et al. manuscript reported an imprinted gradient Zn anode (noted as PVDF-Sn@Zn), where the PVDF-Sn@Zn gradient electrode showed a cycling of 1200 h at low current density/capacity of 1 mA cm⁻²/1 mAh cm⁻² and 200 h at high current density/capacity of 10 mA cm⁻²/10 mAh cm⁻². However, when coupled with the MnO₂@C electrode as a full cell, the cycling performance of the PVDF-Sn@Zn was only slightly better compared with that of Sn@Zn. Moreover, the cycling stability of Sn@Zn||Sn@Zn pouch cell, PVDF@Zn||PVDF@Zn pouch cell, MnO₂@C||Sn@Zn pouch cell and MnO₂@C||PVDF@Zn pouch cell had not been provided. I cannot recommend this work for publication in Nat. Commun. at this stage. Here are my specific points:

____ Thank you very much for the positive support to our manuscript and the valuable suggestions. We have carefully revised our manuscript according to your comments, and the detailed answers are as follows:

Comments 1:

In Line 143-144, the authors stated that the thickness of the upper PVDF layer is about 2.4 μm and the Sn layer is about 4 μm. It is unclear that if these reported values of thickness were optimized parameters for the performance or not. Why did the authors set up the thickness of PVDF layer and Sn layer to be these values? I would suggest the authors to provide additional experimental results to optimize the setup.

____ We thank the reviewer for the valuable comment. The PVDF thickness is optimized by comparing the cycling performance of different PVDF@Zn electrodes with different PVDF thicknesses. PVDF@Zn-5 μm electrode with a PVDF thickness of 5 μm is prepared as shown in Fig. R10. The cycling performance shows that the PVDF@Zn-5 μm//PVDF@Zn-5 μm cell exhibits a kinetic short circuit after only several cycles, much worse than the PVDF@Zn-2.4 μm//PVDF@Zn-2.4 μm cell, which can be attributed to the increased Zn²⁺ ion transfer resistance caused by the unsatisfactory ionic conductivity of PVDF. It can be inferred that PVDF can effectively improve the corrosion resistance of the electrode, but a too-thick PVDF layer is not conducive to uniform Zn deposition on the electrode surface.

The optimization of the thickness of the Sn layer is carried out by controlling the immersion time of the Zn foil in the SnCl₄ solution. As shown in Fig. R6a and b. Sn@Zn-30 s electrode with Sn thickness of 0.7 μm is prepared by placing Zn foil in SnCl₄ solution for 30 s. However, the cycling performance

shows that the Sn@Zn-30 s//Sn@Zn-30 s cell experiences a short circuit after only about 20 h, which can be attributed to the limited modulation property of the thinner Sn layer. We also obtained a thicker Sn layer by soaking the Zn foil in SnCl₄ solution for 10 min, as shown in Fig. R6c and d. The Sn@Zn-10 min electrode shows a porous and non-uniform structure due to severe etching/reaction, and the Sn@Zn-10 min//Sn@Zn-10 min cell experiences a short circuit after only about 16 h.

Based on the above discussion, it is favorable to achieve good cycling stability of the gradient electrode when the thickness of PVDF is 2.4 μm , and the thickness of the Sn layer is 4 μm .

Fig. R10 a Voltage profiles of PVDF@Zn-5 μm //PVDF@Zn-5 μm symmetric cells at current densities/capacities of 2 mA cm⁻²/1 mAh cm⁻². **b** Cross-sectional SEM image of PVDF@Zn-5 μm . Scale bar, 20 μm for **b**.

Fig. R6 a Voltage profiles of Sn@Zn-30 s//Sn@Zn-30 s symmetric cell at current densities/capacities of 5 mA cm⁻²/5 mAh cm⁻². **b** Cross-sectional SEM image of Sn@Zn-30 s. **c** Voltage profiles of Sn@Zn-10 min//Sn@Zn-10 min symmetric cell at current densities/capacities of 5 mA cm⁻²/5 mAh cm⁻². **d** Cross-sectional SEM image of Sn@Zn-10 min. Scale bar, 2 μm for **b** and 5 μm for **d**.

Comments 2:

In Line 215-216, the authors stated that in Fig. 5d, PVDF-Sn@Zn||PVDF-Sn@Zn cell can maintain stable small voltage hysteresis (less than 15 mV for the first 600 cycles) for over 1200 h. However, the voltage hysteresis for PVDF-Sn@Zn in Fig. 5d varied significantly with the testing time. Why was that? In addition, I would suggest the authors to compare the voltage hysteresis after long cycling tests of this work with that of other works reported previously.

_____ We thank the reviewer for the valuable comment. Figure 5d shows the Voltage profiles of Zn//Zn symmetric cells at current densities/capacities of 1 mA cm⁻²/1 mAh cm⁻². It can be obtained from the curve that the voltage has small fluctuations and gradually increases with time. The fluctuation of the voltage could be derived from the changes in the test environment (such as room temperature fluctuations and inner heat generated during continues cycling), which can affect the kinetics of the Zn²⁺/Zn reaction. The gradual increase in voltage can be also attributed to the generation of inert Zn on the electrode surface, which can reduce the Zn²⁺/Zn reaction kinetics.

For comparison, we extracted the voltage hysteresis of the PVDF-Sn@Zn gradient electrode at the cycling of 500th (14.1 mV) and 1000th (19.4 mV) and compared them with the reported values of advanced Zn anodes. As shown in Table R1, the PVDF-Sn@Zn gradient electrode exhibits promising voltage hysteresis during long cycling.

We have also added the discussions in Supplementary Information.

Table R1 Comparison of voltage hysteresis of reported advanced Zn anodes with this work at the current density/capacity of 1 mA cm⁻²/1 mAh cm⁻².

Zn anode	Voltage hysteresis (mV)	Cycle	Ref.	Published Year
A-Zn	~25	500	ACS nano ¹	2022
Zn@ZnF ₂	35.7	800	Adv. Mater. ²	2021
ZnOHF@Zn	27.2	700	Energy Storage Mater. ³	2022
Zn@ZnO HPA-2.0	39.3	400	Adv. Funct. Mater. ⁴	2020
ZnSn-1	~50	100	Adv. Funct. Mater. ⁵	2021
Zn-P-MIEC	> 100	initial	Adv. Mater. ⁶	2022
NGO@Zn	17	20	Adv. Mater. ⁷	2021
ZF@F-TiO ₂	30	100	Nat. Commun. ⁸	2020
Zn _{0.73} Al _{0.27} @Zn	48.3	initial	Nano Lett. ⁹	2022
GFA-5	76	initial	Energy Environ. Sci. ¹⁰	2022
PVDF-Sn@Zn	14.1	500	This work	
	19.4	1000		

References

- 1 Yan, Y. *et al.* Surface-Preferred Crystal Plane Growth Enabled by Underpotential Deposited Monolayer toward Dendrite-Free Zinc Anode. *ACS Nano* **16**, 9150-9162 (2022).
- 2 Yang, Y. *et al.* Synergistic Manipulation of Zn²⁺ Ion Flux and Desolvation Effect Enabled by Anodic Growth of a 3D ZnF₂ Matrix for Long-Lifespan and Dendrite-Free Zn Metal Anodes. *Adv. Mater.* **33**, e2007388 (2021).
- 3 Pan, Z. *et al.* Zincophilic 3D ZnOHF Nanowire Arrays with Ordered and Continuous Zn²⁺ Ion Modulation Layer Enable Long-term Stable Zn Metal Anodes. *Energy Storage Mater.* **50**, 435-443 (2022).
- 4 Kim, J. Y., Liu, G., Shim, G. Y., Kim, H. & Lee, J. K. Functionalized Zn@ZnO Hexagonal Pyramid Array for Dendrite-Free and Ultrastable Zinc Metal Anodes. *Adv. Funct. Mater.* **30**, 2004210 (2020).

- 5 Wang, L. *et al.* Sn Alloying to Inhibit Hydrogen Evolution of Zn Metal Anode in Rechargeable Aqueous Batteries. *Adv. Funct. Mater.* **32**, 2108533 (2021).
- 6 Zhang, M. *et al.* Construction of mixed ionic-electronic conducting scaffolds in Zn powder: A scalable route to dendrite-free and flexible Zn anodes. *Adv. Mater.* **34**, e2200860 (2022).
- 7 Zhou, J. *et al.* Ultrathin Surface Coating of Nitrogen-Doped Graphene Enables Stable Zinc Anodes for Aqueous Zinc-Ion Batteries. *Adv. Mater.* **33**, e2101649 (2021).
- 8 Zhang, Q. *et al.* Revealing the role of crystal orientation of protective layers for stable zinc anode. *Nat. Commun.* **11**, 3961 (2020).
- 9 Zheng, J. *et al.* Electrostatic Shielding Regulation of Magnetron Sputtered Al-Based Alloy Protective Coatings Enables Highly Reversible Zinc Anodes. *Nano Lett.* **22**, 1017-1023 (2022).
- 10 Liang, G. *et al.* Gradient fluorinated alloy to enable highly reversible Zn-metal anode chemistry. *Energy Environ. Sci.* **15**, 1086-1096 (2022).

Comments 3:

In Fig. 5e, the voltage hysteresis for PVDF-Sn@Zn turned smaller at 300-400 h compared with that at 0-300 h (there was a similar phenomenon in Fig. 5f). What were the possible reasons? Please discuss it.

____ We thank the reviewer for the valuable comment. The voltage hysteresis of PVDF-Sn@Zn//PVDF-Sn@Zn cell becomes smaller at 300-400 h (from 27mV to 23 mV), which can be related to changes in the electrode/electrolyte interface and the temperature of the test environment. 1) Changes in the interfacial properties of the metallic Zn and Sn layers after long-term cycling, as well as the re-exposure of the 'dead Zn'-covered Sn layer, can lead to Zn/Zn²⁺ redox kinetics improvement, resulting in a reduction in voltage hysteresis. 2) In addition, changes in the temperature of the test environment can also have a significant effect on voltage hysteresis. To explore the effect of temperature on voltage hysteresis, the voltage-time curves of PVDF-Sn@Zn//PVDF-Sn@Zn cell at different temperatures were investigated. As shown in Fig. R11, PVDF-Sn@Zn//PVDF-Sn@Zn cell exhibits voltage hysteresis of approximately 85.5 mV, 29.5 mV and 17.0 mV at 3°C, 25°C and 50°C, respectively, demonstrating that increased ambient temperature may be a significant contributor to reduced voltage hysteresis.

Fig. R11 Voltage curve for PVDF-Sn@Zn//PVDF-Sn@Zn symmetric cell with current density/capacity of 5 mA cm⁻²/5 mAh cm⁻² at different temperatures.

Comments 4:

In Fig. 5f, the whole testing duration was 200 h; nevertheless, the voltage hysteresis for PVDF-Sn@Zn only turned stable after 160 h. I would suggest the authors to extend the testing hours at 10 mA cm⁻². Moreover, how about the voltage profiles of PVDF-Sn@Zn||PVDF-Sn@Zn cell at larger current densities/capacities?

____ We thank the reviewer for the valuable suggestion. As mentioned above, the reduction in voltage hysteresis in PVDF-Sn@Zn//PVDF-Sn@Zn cells may be related to changes in the electrode/electrolyte interface and the temperature of the test environment. We zoomed in the voltage-time curve of the PVDF-Sn@Zn||PVDF-Sn@Zn cell at 140-190 h, as shown in Fig. R12a. It can be found that the PVDF-Sn@Zn//PVDF-Sn@Zn cell can maintain good Zn plating/stripping kinetics during this period.

Fig. R12b shows the extended voltage-time profile of the PVDF-Sn@Zn||PVDF-Sn@Zn cell at a current density/capacity of 10 mA cm⁻²/10 mAh cm⁻², where the PVDF-Sn@Zn||PVDF-Sn@Zn cell experienced a kinetic short circuit at 216 h, which may be due to the accumulation of 'dead zinc'.

We tested the cycling performance of the PVDF-Sn@Zn||PVDF-Sn@Zn cell at higher density/capacity (15 mA cm⁻²/15 mAh cm⁻²). However, due to the high Zn deposition capacity, it is difficult to maintain the good Zn plating/stripping reversibility, and the PVDF-Sn@Zn||PVDF-Sn@Zn cell develops a short circuit after only short cycles (Fig. R12c).

Fig. R12 a,b Voltage curve for PVDF-Sn@Zn//PVDF-Sn@Zn symmetric cell with current density/capacity of 10 mA cm⁻²/10 mAh cm⁻². **c** Voltage curve for PVDF-Sn@Zn//PVDF-Sn@Zn symmetric cell with current density/capacity of 15 mA cm⁻²/15 mAh cm⁻².

Comments 5:

In Fig. 6d, it seems that the capacity for PVDF-Sn@Zn was just slightly higher than that for Sn@Zn. Does this mean that the role of PVDF in PVDF-Sn@Zn is minor? Moreover, the long-term cycling stability after 630 cycles was missing. However, the authors stated in Line 250-251, initial capacity after 630 cycles, which is much better than the other three cells. This is less convincing.

____ We thank the reviewer for the valuable suggestion. In this work, the role of PVDF in PVDF-Sn@Zn is mainly contributed to enhanced cycling stability. The higher capacity exhibited by the PVDF-Sn@Zn gradient electrode compared to the Sn@Zn electrode can be attributed to the microchannels induced by the imprinting process, which optimized Zn²⁺ ion flux and increased specific surface area. More importantly, the top PVDF layer and the bottom Sn layer creates gradient conductivity and hydrophilicity in the electrode, which induces the preferential deposition of Zn in the microchannels. As a result, the PVDF-Sn@Zn gradient electrode can exhibit much better Zn plating/stripping kinetics and long-term cycling performance.

To further demonstrate the good cycle stability of the PVDF-Sn@Zn gradient electrode, we extended the cycle performance of the full cells to 700 cycles. As shown in Fig. R13, the MnO₂@C//PVDF-Sn@Zn cell can retain 70.3% of its initial capacity in the 700th cycle, obviously better than the other three cells (MnO₂@C//Sn@Zn-53.3%, MnO₂@C//PVDF@Zn-43.4% and MnO₂@C//Zn foil-failed). We have revised accordingly in the manuscript.

Fig. R13 Long-term cycling stability at a current density 2 A g⁻¹ of different full cells.

Comments 6:

In Fig. 6f-h, please provide the corresponding results of PVDF@Zn and Sn@Zn.

____ We thank the reviewer for the valuable comment. The cycling performance of the symmetric pouch cell, the CV curve and the cycling performance of the full pouch cell based on the PVDF@Zn and Sn@Zn electrode are shown in Fig. R14. As shown in Fig. R14a, the PVDF-Sn@Zn//PVDF-Sn@Zn pouch cell maintains a small voltage hysteresis and stable cycling for more than 500 h at a current density/capacity of 2 mA cm⁻² (40 mA)/1 mAh cm⁻² (20 mAh), better than Sn@Zn//Sn@Zn (231 h), PVDF@Zn//PVDF@Zn (63 h) and Zn foil//Zn foil (36 h) pouch cells. MnO₂@C//PVDF-Sn@Zn pouch cell also exhibits superior reaction kinetics with lower voltage polarization than that for other three pouch cells (Fig. R14b). The stability test reveals that the MnO₂@C//PVDF-Sn@Zn pouch cell can retain more than 65.5% of the initial capacity in the 250th cycle, which is also much better than the value for MnO₂@C//Sn@Zn (49.3%), MnO₂@C//PVDF@Zn (39.1%) and MnO₂@C//Zn foil (32.3%) pouch cells (Fig. R14c). We have revised accordingly in the manuscript.

Fig. R14 **a** Stability of symmetric pouch cells at a current density of 2 mA cm^{-2} (40 mA) with a capacity of 1 mAh cm^{-2} (20 mAh). **b** CV curves and **c** cycling stability of full pouch cells at a current density of 2 A g^{-1} .

Comments 7:

It would be helpful if the authors could provide theoretical calculations comparing PVDF-Sn@Zn with Zn foil, PVDF@Zn and Sn@Zn to validate the advantages of the PVDF-Sn@Zn gradient anode.

_____ We thank the reviewer for the valuable suggestion. To reveal the advantages of the PVDF-Sn@Zn gradient anode over the other three controlled electrodes, theoretical calculations of the current density and Zn^{2+} ion concentration distribution are studied in Fig. R15.

Current density distribution indicates that the roughness of Zn foil would result in concentrated current density on the bumps, which causes severe polarization (Fig. R15a). The non-conductivity of the PVDF layer can effectively reduce the current density on the electrode surface, but it is still uneven in these bumps (Fig. R15b). The Sn layer with good electrical conductivity also cannot solve the problem, with current density accumulated in bumps and resulting in uneven Zn deposition (Fig. R15c). In comparison, the non-conductive PVDF layer on the top of the PVDF-Sn@Zn gradient electrode can effectively reduce the current density at the top surface of the electrode, thereby inducing the preferential deposition of Zn in the imprinted microchannels and improving the cycle performance (Fig. R15d).

The Zn^{2+} ion concentration distribution shows that the Zn^{2+} ion concentration on the Zn foil is low due to its unsatisfactory Zn affinity and high nucleation overpotential (Fig. R15e). The non-conductive

and hydrophobic PVDF will further reduce the Zn^{2+} ion concentration on the surface of Zn metal (Fig. R15f). The Sn layer can increase the Zn^{2+} ion concentration of the Zn/electrolyte interface due to its good zincophilicity and hydrophilicity, thus can effectively improve the Zn deposition uniformity (Fig. R15g). PVDF-Sn@Zn gradient electrode benefits from the advantages of the 3D structure, which can accelerate the diffusion of Zn^{2+} ions and increase the Zn ion concentration of the electrode surface. In addition, the gradient design induces the Zn^{2+} ion concentration incline to microchannels, resulting in uniform deposition of Zn in the microchannels (Fig. R15h).

We have also added the discussions in Supplementary Information.

Fig. R15 Current density simulation on **a** Zn foil, **b** PVDF@Zn, **c** Sn@Zn and **d** PVDF-Sn@Zn gradient electrodes. Zn^{2+} ion concentration distribution simulation on **e** Zn foil, **f** PVDF@Zn, **g** Sn@Zn and **h** PVDF-Sn@Zn gradient electrodes.

REVIEWERS' COMMENTS

Reviewer #1 (Remarks to the Author):

The authors have addressed the comments of reviewers, the quality of the paper was correspondingly improved, it can be recommended for publication as is.

Reviewer #2 (Remarks to the Author):

The authors have adequately addressed my concerns with the original draft, as well as those of the other reviewers, and have noticeably improved the quality of the work as a result. This manuscript is now suitable for publication in Nature Communications.

Reviewer #3 (Remarks to the Author):

In the revised manuscript, the authors have addressed most of the comments/concerns raised by this reviewer properly, and the revised form has been nicely improved. However, there is still an issue I would like to recommend the authors to address further before acceptance. Upon Comments #1 of my previous round of evaluation, the authors provided additional experiments and stated that the thickness of the Sn layer used was an optimized parameter. However, the authors only provided the results of the PVDF layer with the thickness of 5 μm as a control group. It would be better if the authors could provide results for the case of the thinner PVDF layer (with a thickness less than 2.4 μm , e.g. 1 μm) to validate that the PVDF layer with the thickness of 2.4 μm was rationally optimized.

Response to reviewer's comments:

Reviewer 1

The authors have addressed the comments of reviewers, the quality of the paper was correspondingly improved, it can be recommended for publication as is.

_____ We appreciate the reviewer's support and recommendation of our work for publication in Nature Communications.

Reviewer 2

The authors have adequately addressed my concerns with the original draft, as well as those of the other reviewers, and have noticeably improved the quality of the work as a result. This manuscript is now suitable for publication in Nature Communications.

_____ We appreciate the reviewer for his/her positive comments and recommendation of our work for publication in Nature Communications.

Reviewer 3

In the revised manuscript, the authors have addressed most of the comments/concerns raised by this reviewer properly, and the revised form has been nicely improved. However, there is still an issue I would like to recommend the authors to address further before acceptance. Upon Comments #1 of my previous round of evaluation, the authors provided additional experiments and stated that the thickness of the Sn layer used was an optimized parameter. However, the authors only provided the results of the PVDF layer with the thickness of 5 μm as a control group. It would be better if the authors could provide results for the case of the thinner PVDF layer (with a thickness less than 2.4 μm , e.g. 1 μm) to validate that the PVDF layer with the thickness of 2.4 μm was rationally optimized.

_____ Thank you very much for the positive support to our manuscript and the valuable suggestions. We have carefully revised our manuscript according to your comments, and the detailed answers are as follows:

Since there is a drying process after PVDF coating, a too thin layer of PVDF would result in cracks and nonuniform coating after the drying procedure. To verify this, a PVDF@Zn-0.7 μm electrode with a PVDF thickness of \sim 0.7 μm is prepared as shown in Fig. R1. The cycling performance shows that the PVDF@Zn-0.7 μm /PVDF@Zn-0.7 μm cell exhibits a short circuit after about 126 h, which can

be attributed to limited improvement in corrosion resistance with the nonuniform coating (certain cracks can be observed from the SEM image).

We have added the above revision in the revised Supplementary Information.

Fig. R1 a Voltage profiles of PVDF@Zn-0.7 μm //PVDF@Zn-0.7 μm symmetric cell at current density/capacity of 2 mA cm⁻²/1 mAh cm⁻². **b** Cross-sectional SEM image of PVDF@Zn-0.7 μm . Scale bar, 3 μm for **b**.